# Space-time variability of soil moisture droughts in the Himalayan region

Santosh Nepal[1], Saurav Pradhananga[1], Narayan Kumar Shrestha[1,2], Sven Kralisch[3,4], Jayandra Shrestha[1,5], Manfred Fink[3]

[1]International Centre for Integrated Mountain Development (ICIMOD), Kathmandu, Nepal
[2] School of Engineering, University of Guelph, Guelph, Canada
[3] Department of Geoinformation Science, Friedrich Schiller University Jena, Jena, Germany
[4] Institute of Data Science, German Aerospace Center (DLR), Jena, Germany
[5] Department of Civil and Natural Resources Engineering, University of Canterbury, Christchurch, New Zealand

*Correspondence to*: Santosh Nepal (Santosh.Nepal@icimod.org)

**Abstract**

Soil water is a major requirement for biomass production and therefore one of the most important factors for agriculture productivity. As agricultural droughts are related to declining soil moisture, this paper examines soil moisture drought in the transboundary Koshi River basin (KRB) in the Central Himalayan region. By applying the J2000 hydrological model, daily spatially distributed soil moisture is derived for the entire basin over a 28-year period, 1980–2007. A multi-site and multi-variable approach – streamflow data at one station and evapotranspiration data at three stations – was used for the calibration and validation of the J2000 model. In order to identify drought conditions based on the simulated soil moisture, the Soil Moisture Deficit Index (SMDI) was then calculated, considering the derivation of actual soil moisture long-term soil moisture on a weekly timescale. To spatially sub-divide the variations in soil moisture, the river basin is partitioned into three distinct geographical regions, trans-Himalaya, the mountains, and the plains. Further, the SMDI is aggregated temporally to four seasons – winter, pre-monsoon, monsoon, and post-monsoon – based on wetness and dryness patterns observed in the study area. This has enabled us to look at the magnitude, extend and duration of soil moisture drought. The results indicated that the J2000 model can simulate the hydrological cycle of the basin with good accuracy. Considerable variation in soil moisture was observed in the three physiographic regions and across the four seasons due to high variation in precipitation and temperature conditions. The year 1992 was the driest year and 1998 was the wettest at the basin scale in both magnitude and duration. Similarly, the year 1992 also has the highest number of weeks under drought. The study has found an increase in frequency in the later years of the period under study, most visibly in the pre-monsoon season. Comparing the SMDI with the standardised precipitation index (SPI) suggested that SMDI can reflect a higher variation of drought conditions than SPI. This calculation is based on a high-resolution spatial representation of soil moisture, which was simulated using a fully distributed hydrological model. Our results suggested that both the occurrence and severity of droughts have increased in the Koshi River basin over the last three decades, especially in the winter and pre-monsoon seasons. The insights provided into the frequency, spatial

coverage, and severity of drought conditions can provide valuable contributions towards an improved management of water resources and greater agricultural productivity in the region.

**Keywords:** Soil moisture deficit index (SMDI), drought, hydrological modelling, J2000 model, standardised precipitation index (SPI), Koshi River basin

## 1. Introduction

Droughts are considered one of the world's major social and economic hazards, which have been increasing in recent decades. Given the central role of agricultural productivity in the economic development of a nation, water resource planners and managers need a system that can assess and forecast different forms of agricultural drought. There are different forms of drought, but they are all linked to a great extent to precipitation and temperature variability (Mishra and Singh, 2010). There are mainly four types of droughts: meteorological, soil moisture, hydrological, and socioeconomic (Van Loon, 2015). Soil moisture drought, also referred as agricultural drought (Mishra and Singh, 2010), is a major concern as it is directly related to agricultural productivity and can have direct and adverse implications for a nation's economy (Sheffield et al., 2004; Wang, et al., 2011a).

Droughts impact both surface and groundwater resources and can lead to reduced water supply, impaired water quality, crop failure, diminished hydropower generation, disturbed riparian habitats, and adversely affect a host of economic and social activities (Riebsame et al., 1991). Understanding the processes that cause droughts and their spatial and temporal variability is thus essential for a sustainable management of natural resources (Stefan et al., 2004). In particular, land and water resources planners need to understand historic drought events, their magnitude and severity, and develop measures to forecast and mitigate the impacts of future droughts. Furthermore, the demand for water resources has increased in recent decades due to growing populations, and increased demands by agriculture and industry. Global warming has further contributed to water scarcity and uncertainties in the availability of water across space and time (Mishra and Singh, 2010; Wang et al., 2011b).

Soil moisture dynamics are a function of atmospheric conditions and the characteristics of soils and vegetation. Whereas an increase in precipitation increases the moisture content in the soil, higher temperatures associated with high wind speed, greater radiation, and low humidity cause dryness due to increased evapotranspiration. Soil moisture droughts are therefore closely related to the land surface water and energy cycles. Changes in soil moisture directly affect water availability, plant productivity, and crop yields. It is clear, therefore, that soil moisture deficits have critical implications for both water supply and agriculture (Wang et al., 2011a).

Closely related to agricultural drought is hydrological drought, which is defined as a period with inadequate surface and subsurface water resources for established water uses. Its frequency and severity are usually assessed on a river basin scale (Mishra and Singh, 2010; Van Loon, 2015). While all droughts originate from a lack of precipitation, hydrological droughts are more concerned with how this deficiency plays out through a hydrological system. Agricultural and hydrological droughts

can be seen as delayed responses to meteorological droughts. It takes a while for deficiencies in precipitation to show up in components of the hydrological system such as soil moisture, streamflow, and groundwater levels (Van Loon, 2015).

There are many drought indices, such as the Standardised Precipitation Index (SPI), Standardised Precipitation Evaporation Index (SPEI), Evapotranspiration Deficit Index (ETDI), Soil Moisture Deficit Index (SMDI), Aggregate Drought Index (ADI), Standardised Runoff Index (SRI), probabilistic precipitation vegetation index (PPVI) and Palmer Drought Severity Index (PDSI), which indicates the differential nature of droughts that might occur at different time intervals and lag times (Bayissa et al., 2018; Huang et al., 2015; Narasimhan and Srinivasan 2005; Monteleone et al. 2020). Focusing on soil moisture,

variability, SMDI takes into account more variables (such as evapotranspiration, soil properties, and root depth) than SPI and SPEI, which takes into account precipitation, and precipitation and evapotranspiration respectively. Therefore, SMDI can provide dependable information to interpret the occurrence and severity of the agricultural drought. Similarly, SPI is a widely used index to characterise meteorological droughts on a range of timescales. Monteleone et al. (2020) suggested list of indices for agriculture drought monitoring, including Evapotranspiration Deficit Index (ETDI), Normalised Difference Vegetation

Index (NDVI), Soil Moisture Anomaly Index (SMAI), SPI, SMDI and Standardised Soil Moisture Index (SSI). Each of these indices has their own pros and cons for different climatic variables they use for drought calculation, data requirement and availability and their potential use for agricultural drought monitoring.

Climate change is very likely to exacerbate different forms of drought and other impacts on various sectors (IPCC, 2014, 2019; Wang et al. 2011b). Studies have suggested an increased incidence of drought over many land areas since the 1950s (Dai,

2011, 2013). Floods and droughts are commonly felt major natural hazards in the Himalayan region. According to a report of the Intergovernmental Panel on Climate Change (IPCC), increases in floods and droughts will exacerbate rural poverty in parts of Asia as a result of negative impacts on the rice crop and resulting increases in food prices and the cost of living (Hijioka et al., 2014). Because of the uneven intra-annual distribution of precipitation in the Central Himalayan region, the region suffers from floods during the monsoon season (June–September), while only 20% of the annual precipitation occurs during the

remaining eight months of the year, causing agricultural production to suffer due to the lack of water. However, in contrast to floods, the literature pertaining to soil moisture drought is limited, particularly for the Himalayan region.

In India, droughts are a regular phenomenon and have impacted various sectors (Prabhakar and Shaw, 2008). Since the mid-1990s, prolonged and widespread droughts have occurred in consecutive years, and the frequency of droughts has also increased in recent times (Mishra and Singh, 2010). Mallya et al. (2016) indicate an increasing trend in the severity and

90 frequency of droughts during recent decades, a trend in which droughts are becoming more regional around the Indo-Gangetic plains, including other areas. In China, trends suggest that soil moisture droughts became more severe, prolonged, and frequent between 1950 and 2006, especially in north-eastern and central China, suggesting an increased susceptibility to agricultural drought (Wang et al., 2011a). According to Su et al. (2018), in China, the estimated losses due to drought under a global average temperature rise of 1.5°C will be ten times higher as compared with the reference period 1986–2005 and nearly

95 threefold relative to 2006–2015.

In Nepal too, a few studies have indicated increasing trends in different forms of drought. In the transboundary Koshi River basin (KRB), Shrestha et al. (2017) presented spatial and temporal trends in historically known drought events using the SPI. Joshi and Dangol (2018) suggested that the severe drought over the last ten years in one of the middle hill districts in the KRB has caused major spring sources and rivers to dry up and compelled the local community to migrate to other areas with better water security. Wu et al. (2019) suggested significant spatial heterogeneity of droughts in the KRB with higher crop water shortage index (CWSI) values in its downstream (the plains of Nepal and India) and upstream regions (parts of China) than midstream (the middle and high mountains). Few studies have indicated warmer and wetter climate in the KRB towards the end of the century (Kaini et al. 2019; Rajbhandari et al. 2015) which might have adverse impacts on soil moisture and related droughts in the future. Hamal et al. 20202 suggested the frequent occurrence of severe drought episodes (1992, 1994, 2006, 2008, 2009, 2012 and 2015) during the cropping cycle of summer maize and winter wheat in Nepal based on the SPEI indices. The results also suggest that drought has affected the crop yield over different regions of Nepal. There are few agricultural droughts reported in the literature on the mountain region of Nepal. Some prominent drought events are the winter drought of 2005–2006 and the summer droughts of 1992 and 2005, which caused a decrease in agricultural production (Bhandari and Panthi, 2014; Dahal et al., 2016; Regmi, 2007).

The transboundary Koshi River basin, which straddles parts of China, Nepal, and India, faces both floods and droughts due to its unique climatic system. The lowland Indo-Gangetic Plain, whose highly fertile lands provide food for millions of people, has a tropical climate dominated by the summer monsoon. The mountainous part in the southern Himalaya is also influenced by the monsoon but the spatial variation is very high due to the orographic effect. It has a temperate to alpine climate. The Tibetan Plateau in the northern part of the KRB has an alpine climate with dry and cold conditions and is less influenced by the monsoon due to high mountain barriers. As such, there is great variability in the spatial distribution of annual precipitation, ranging from 500 millimetres (mm) in the northern KRB to over 4,500 mm in southern Nepal (Karki et al., 2016).

This paper aims at assessing soil moisture droughts in the Koshi River Basin. To understand soil moisture droughts, this paper considered the Soil Moisture Deficit Index (SMDI) and Standardised Precipitation Index (SPI) for 28 years (1980–2007). For soil moisture variability, SMDI takes into account precipitation, temperature, evaporation, soil and vegetation properties affecting soil moisture conditions. For this purpose, the basin's soil moisture was simulated with the use of the process-based J2000 hydrological model, which was validated against observed discharge and evapotranspiration. The J2000 model has been successfully used to investigate hydrological droughts in Central Vietnam (Firoz et al., 2018; Nauditt et al., 2017). This paper specifically investigates the spatial and temporal variability of soil moisture for the trans-Himalaya (Tibet), the high and middle mountains (Nepal), and the southern plains of the river basin (in Nepal and India). We also compared the SMDI with the SPI to identify the variation of the drought indication in space and time. To the best of our knowledge, soil moisture drought is being studied for the first time in the Central Himalaya region and this paper provides insights into its spatio-temporal variability in the historic time period under consideration.

**2 Study Area**

The Koshi is a major tributary of the Ganges River. The transboundary Koshi River basin (KRB) is located in the Central
Himalaya. The world's highest peak, Mt Everest (8,848.86 m a.s.l.), and the world's third-highest mountain, Mt Kanchenjunga (8,586 m a.s.l.), are located in the KRB (Figure 1). The river drains a region extending from the trans-Himalaya (the northern slopes of the Himalaya in China) to the southern slopes of Nepal and flows to the Indo-Gangetic Plain in India. Its total catchment area is 87,570 km$^2$ at its confluence with the Ganges River in Kursela, India (Figure 1). It covers much of eastern Nepal barring the Mai-Kankai River basin which originates from the Siwalik Hills of Nepal (Figure 1, inset map). Chatara is
a gauging station where the model has been calibrated and validated, covering about two-thirds (about 58,000 km$^2$) of the basin's area.

Based on topography, the KRB is divided into five physiographic regions. The Terai region in the south is a low-lying plains area (60–300 m asl). The Siwalik region is a narrow, foothill belt with an elevation of 300–1,000 m asl, while the middle mountain region, with steep slopes and deep-cut valleys, is the widest strip, with elevations of 1,000–3,000 m asl. The high
mountain region, with elevations above 3,000 m asl, is to the north and generally above the snow line (Dhital, 2015). The trans-Himalaya represents the Tibetan Plateau, located in China. This study investigates drought conditions in three regions, which will be referred to as trans-Himalaya, mountains, and plains in the sections that follow (Figure 1):

-    **Trans-Himalaya**: the part of the KRB in the trans-Himalaya that falls in Tibet, China;
-    **Mountains**: the high and middle mountains (north of the Terai up to the Himalayan peaks in Nepal); and
-    **Plains**: the Terai in Nepal and the Indo-Gangetic Plain in India.

**3. Methodological Approach**

This section introduces the J2000 hydrological model, presents the model input data, describes the modelling strategy, and the calculation of the SMDI and SPI.

**3.1 The J2000 hydrological model**

J2000 is a modular, spatially distributed, process-oriented hydrological model developed inside the JAMS modelling system (Kralisch and Krause, 2006; Krause, 2001). The JAMS framework allows building hydrological models by combining individual modelling components in a very flexible way. Existing JAMS models such as J2000 can therefore easily be adapted to address specific problems. Moreover, JAMS provides several functions that are often required during the development of hydrological models and application workflows, for example, for analysing model results or for performing model calibration
or sensitivity analyses (Krause et al., 2009). To support more complex data processing tasks that typically occur when processing large datasets or during model calibration, the framework provides parallel computing functions (Kralisch and

Fischer, 2012) and service-based simulations on remote computer servers. The J2000 model has been widely used in river catchments around the globe including in the Himalayan region (Eeckman et al., 2019; Nepal et al., 2017; Shrestha and Nepal, 2019).

The J2000 model comprises modules to represent all important hydrological processes. A short description of the main process simulation modules is provided below. All of them contain some calibration parameters that have to be adapted during the application of the model. A detailed description of these parameters, and the modules to which they are related, are provided in Nepal (2012). To represent hydrological processes within the watershed in a spatially distributed way, the spatial discretization concept of hydrological response units (HRUs) (Flügel, 1995) is used to delineate modelling entities. It will be

described in section 3.2. The second type of modelling entity in the J2000 model is river segments (reaches), which are used to represent water transport in the river bed. The model uses a fixed temporal resolution of daily time steps. Accordingly, the hydrological process simulation is performed at each time step and for each HRU. It can be summarised in the following way. In a first step, climate input parameters that are provided as point data (for example, measurements at climate stations) are interpolated such that a local value is generated for each HRU and time step. These climate parameters include min/mean/max

temperature, precipitation, humidity, sunshine duration, and wind speed.

The distribution of precipitation between rain and snow is simulated depending on the air temperature. To determine the amount of rain and snow, it is assumed that temperatures below a certain threshold result in precipitation entirely as snow and those exceeding a second threshold results entirely in rainfall. The interception module uses a simple storage approach and assumes a maximum interception storage capacity based on the leaf area index (LAI) of the respective land cover. The snow

module calculates the different phases of snow accumulation, metamorphosis, and snowmelt. Snowmelt depends on the energy input provided by the air temperature, and the soil heat flux, and is considered as the potential melt rate. The snowpack can store liquid water in its pores up to a certain critical density. In the model, the snowmelt runoff from the snowpack is passed to the soil module through infiltration. The antecedent soil moisture conditions influence the rate of infiltration (Krause, 2001). In the glacier area, the same snowmelt process is applied. The glaciated area is divided into clean and debris-covered glaciers,

based on slope and elevation. In the case of the KRB, glaciers at an elevation above 4,500 m a.s.l. and with slopes greater than 15 degrees are considered as clean. Once the seasonal snow cover melts, glacier ice melt starts. This is estimated based on an enhanced degree-day factor approach which takes into account temperature, radiation, and whether the glacier is clean or debris-covered. Rainfall on the glacier's surface is also taken into account. The run-off from the glacier area is separated into the components snowmelt, glacier ice melt, and rain run-off. All of them are then routed to the next stream in the reach networks

(Nepal et al., 2014). The potential evapotranspiration is calculated according to the Penman-Monteith approach (Allen et al., 1998). This approach considers the meteorological input regionalized for each HRU in the first step to calculate the potential evapotranspiration.

The central and most complex component of the J2000 model is the soil water module, which controls the regulation and distribution of the consecutive water fluxes. The soil zone of each HRU is subdivided into two storages according to the

specific pore volumes of the soil. Middle pore storage (MPS) represents the pores with a diameter of 0.2–50 µm, in which water is held against gravity but can be reduced by plant transpiration as part of the evapotranspiration process. Therefore, in the J2000 model context, soil moisture is considered up to the depth at which plant root depth can affect the availability of soil moisture. It is therefore different for different land cover types. The MPS thus represents the usable field capacity in the model. Large pore storage (LPS) represents the pores with a diameter of more than 50 µm. These cannot hold water against gravity

and provide the water fluxes for the subsequent compartments and run-off components using calibrated delay functions. The direct rainfall and other water inputs (for example, from snowmelt) can provide inputs to the soil water through the infiltration process. Water in the LPS is distributed into lateral components (outflow as interflow) and vertical components (outflow as percolation), depending on the slope. Water storage will be depleted by the actual evapotranspiration, which is limited by the potential evapotranspiration and the actual water saturation of the MPS (Krause, 2001).

For the SMDI calculations, this study considered soil moisture as the water which is stored in the MPS. The LPS was not considered because the water in large pores is not used in evapotranspiration directly (only by diffusion to the MPS) and leaches out of the soil. The water inputs for the soil module are from snowmelt, rainfall, and lateral fluxes from HRUs located upstream. First, infiltration is calculated by an empirical approach, based on actual soil moisture and the maximum infiltration parameter differentiated in summer, winter, and snow cover situations. Any water not able to infiltrate is stored at the surface

in a depression storage up to a certain amount, and any surplus is treated as surface runoff and routed to the adjacent downstream HRU or river reach. Infiltrated water is distributed between the MPS and LPS depending on the actual water saturation of these storages. The percolation is conveyed to the groundwater module. The interflow is routed to the next HRU or river reach.

       The groundwater module of the J2000 model follows a simple storage concept, which contains two groundwater storages for

each HRU. The storage in the upper groundwater zone can be considered as the weathered layer on top of bedrock (Supplementary Figure 1). Similarly, the storage in the lower groundwater zone represents saturated groundwater aquifers. The input from percolation is distributed between the two storages depending on the slope of the model unit and a distribution parameter. The calculation of water discharge from the two storages is done according to the current storage amounts in the form of a linear outflow function using storage retention coefficients for the two storages.

The J2000 model features two routing modules. The lateral routing between HRUs describes water transfers within a flow cascade from one HRU to another from the upper catchment areas until it reaches a stream. The second routing module simulates flow processes in a stream channel by using the commonly applied kinematic wave approach and the calculation of velocity according to Manning and Strickler (Krause, 2001). The only model parameter that needs to be estimated by the user is a routing coefficient, which influences the travel time of the water within a reach. In addition to the water transport within

a reach, the routing module also simulates the water transfer to the adjacent downstream river reach until it reaches the catchment outlet.

## 3.2 Model input data

The J2000 model uses a representation of the catchment and its distributed hydrological characteristics based on hydrological response units (HRUs) (Flügel 1995). The Digital Elevation Model (DEM), and the land use, soil, and geology maps were analysed and combined in an overlay analysis to derive the HRUs (Supplementary Table 1). Altogether, 18,557 HRUs were delineated within the Koshi River basin with an average size of the HRUs as 4.7 km². The HRUs were further separated into three regions (trans-Himalaya, mountains, and plains) for a spatially separated analysis of modelling results and SMDI calculations.

The discharge data and potential evapotranspiration (PET) data used to validate the model were acquired from the Department of Hydrology and Meteorology, Nepal (DHM). In addition, precipitation datasets of the Asian Precipitation – Highly-Resolved Observational Data Integration Towards Evaluation of Water Resources (APHRODITE) (V1101)   project and temperature datasets of the Climate Forecast System Reanalysis (CFSR) project were used for the data-scarce trans-Himalaya region of the KRB. For the lowland plains of the river basin, datasets of the Indian Meteorological Department (IMD) were used for both precipitation and temperature. For the portion of the basin in Nepal, meteorological input datasets (pertaining to precipitation, temperature, relative humidity, wind, and sunshine hours) acquired from the DHM were used in the model. The number of stations for different climate variables are provided in Supplementary Table 1.

Based on the datasets used, Supplementary Table 2 shows the average monthly and annual precipitation and temperature for the period 1980–2007 of three physiographic regions of the KRB. The high and middle mountains get the highest annual precipitation (~2,100 mm) while the trans-Himalaya gets the lowest (~575 mm) with the plains (~1,600 mm) in between. Much of the precipitation in all these three regions falls during the monsoon season (June–September). The average monthly temperature differs drastically between the three regions. Trans-Himalaya exhibits the highest temperature variation with a high of 7ºC during July (summer season) and a low of −13ºC during January (winter). The plains have the highest temperature for each month as it has the lowest elevation (30–280 m). The average monthly temperature varies from 16ºC to 30ºC in the plains. The average monthly temperature for the high and middle mountains has a range of 7ºC–20ºC (Figure 2).

## 3.3 Hydrological modelling

### Calibration and validation

The J2000 hydrological model was applied daily between 1979 and 2007. Since the PET is not calibrated in J2000, the model was validated with PET data from three locations (Kathmandu, Okhaldhunga, and Jiri) in the mountains of Nepal first (Figure 1). Measured PET data at these locations were compared with PET data from the model. After this, the model was manually calibrated and validated with the discharge data at Chatara. We have taken base parameter sets from the Dudh Koshi River basin from Nepal et al. (2014). Nepal et al. (2017) also showed the spatial transferability of parameters from Dudh Koshi to Tamor catchment within the Koshi River basin. Similarly, Eeckman et al. (2019) also used Dudh Koshi parameters for micro

catchments (~5 km$^2$) within the Dudh Koshi basin and suggested that the parameters related to groundwater, surface run-off coefficient, and percolation may change due to the scale of the watershed. In this study as well, a few parameters such as surface and groundwater recession, percolation, and flood routing were changed to match the discharge response for the Koshi basin (Supplementary Table 3). The time period 1985–1995 was used for calibration and 1996–2007 for validation. Due to the unavailability of discharge data from the Indian part of the river basin, the model was first calibrated and validated with the discharge data at Chatara, Nepal (Figure 1). After calibration and validation at Chatara, the model parameters were applied to the whole river basin (including those parts in India, area 87,530 km$^2$) to simulate the required variables of the downstream part of the Koshi. The results are compared with four efficiency criteria, namely, the NSE: Nash Sutcliffe Efficiency (Nash and Sutcliffe, 1970); KGE: Kling Gupta Efficiency (Gupta et al., 2009); $R^2$ (coefficient of determination); and PBIAS (percentage bias). Based on the calibration and validation, soil moisture analysis was conducted for the period 1979–2007 (with 1979 as a warm-up period).

The soil moisture derived from the J2000 model could not be validated directly due to the lack of observed soil moisture data in the basin. While most of the remote sensing-based soil moisture is available only after 2015 (see e.g. Alemohammad et al. 2018), very few like the Climate Change Initiative Soil Moisture product (CCI SM) by European Space Agency (ESA) is available at 25 x 25 km resolution from 1978 to 2015 (Dorigo et al. 2017). Besides, these products differ in considered soil depth when compared to the J2000 model. The spatial resolution of the J2000 model is based on hydrological response units (HRUs) of an average size of 4.7 km2, whereas all available satellite-based soil moisture products feature a distinctly lower spatial resolution. As an example, the CCI SM product has a spatial resolution of 625 km2. Also, remote sensing products might capture artificial water storage, surface irrigation and snow cover, which also affect the spatial and temporal patterns of soil moisture. Because of these differences along with the J2000 model-derived soil moisture which typically considers root depth of vegetation which can reach up to 100 cm soil depth, direct comparison with satellite-derived soil moisture would not be reasonable in this study. However, a monthly comparison with CCI SM is provided in Supplementary Figure 3 and discussed in the 'Discussion' section.

### 3.4 Drought Indices

### 3.4.1 Soil Moisture Deficit Index

The Soil Moisture Deficit Index (SMDI), developed by Narasimhan and Srinivasan (2005), accounts for variability in soil moisture over a long period. Soil moisture can be derived from hydrological models as an intermediate result along with other components of the hydrological cycle (for instance, discharge). Soil moisture is one of the most important parameters in assessing agricultural drought, and the number of SMDI applications to assess it has grown in recent years.

The SMDI was developed with three major characteristics: the ability to assess short-term dry conditions, the ability to indicate drought in any season, and the ability to function in any climatic zone. These characteristics of the SMDI are ideal for agricultural drought monitoring (Narasimhan and Srinivasan, 2005). The calculation of the SMDI involves the calculation of the soil-water deficit (SD) from soil water/moisture (SW). An average weekly soil moisture product can be used as an indicator of short-term drought, depending upon the availability of soil moisture data at different depths or in a lumped way. The J2000 hydrological model computes soil moisture in the root zone of the soil profile. This is a useful index for identifying and monitoring droughts affecting agriculture. The SMDI has a value between –4 (extremely dry) to +4 (extremely wet) and is derived as defined in Equation 1.

The SMDI is categorised as extremely wet (+4 to +3), severely wet (+3 to +2), moderately wet (+2 to +1), normal (+1 to 1), moderately dry (–1 to –2), severely dry (–2 to –3), and extremely dry (–3 to –4), which reflect the range of soil moisture conditions. The equation for the calculation of the weekly SMDI is presented below:

$$SMDI_{y,w} = 0.5 \times SMDI_{y,w-1} + \frac{SD_{y,w}}{50} \tag{1}$$

where

$$SD_{y,w} = \frac{SW_{y,w} - MSW_w}{MSW_w - \min SW_w} \times 100, \qquad if\ SW_{y,w} \leq MSW_w$$

$$SD_{y,w} = \frac{SW_{y,w} - MSW_w}{\max SW_w - MSW_w} \times 100, \qquad if\ SW_{y,w} > MSW_w$$

Where, w indicates week and y indicates year; SD = soil water deficit; MSW, min (SW), and max (SW) = median, minimum, and maximum soil water, respectively.

The calculation of the SMDI has been implemented in the JAMS modelling system using two individual JAMS components, namely *SMDI_DataCollect* and *SMDI_Calc*. The first component is used to collect soil moisture data for each HRU during the normal hydrological simulation with J2000. In addition, this component also calculates long-term soil water statistics for each HRU (for example, $MSW_w$). Once this is finished, the second component (*SMDI_Calc*) will calculate the SMDI values for each HRU based on their weekly soil moisture values ($SW_{y,w}$) and long-term statistics ($MSW_w$, $minSW_w$, $maxSW_w$). While weekly intervals are used as the default, the component can calculate SMDI values based on any given aggregation period, for example, to consider individual characteristics of specific vegetation types. As described above, the HRUs were segregated into three geographical regions, trans-Himalaya, mountains, and plains, as the climatic conditions are different in each of these zones. Similarly, the SMDI values were analysed separately for four seasons: monsoon (June–September), post-monsoon (October–November), winter (December–February), and pre-monsoon (March–May). Since these seasons are defined based on variations in precipitation and temperature, the SMDI is calculated for these seasons to track the variation caused by these meteorological drivers. For this, we averaged the weekly SMDI values for a given season. In this way, the dominating climatic characteristics are maintained at the seasonal level.

### 3.4.2 Standardised Precipitation Index

The Standardised Precipitation Index (SPI) is the most commonly used indicator for detecting and characterising
meteorological drought on different timescales. We calculated the seasonal SPI which was implemented as a JAMS
component. The SPI is calculated based on a long-time series of precipitation data. The SPI measures precipitation anomalies
based on a comparison of observed total precipitation amounts for an accumulation period (for example, 1, 3, 12, or 48 months)
with the long-term historic record for that period. The probability distribution of the historic record was fitted to a gamma
distribution, which was then transferred to a normal distribution to get a mean SPI value of zero (McKee et al., 1993; McKee
et al., 1995). To compare the seasonal SMDI with the SPI, we calculated the SPI data for the same period of four seasons used
to calculate the SMDI. For this, the aggregation period was based on the end month of each season and SPI accumulation
period was chosen based on the months. For winter, 3 months SPI was calculated for the month of February; for pre-monsoon,
3 months SPI for May; for monsoon, 4 months SPI for September and for post-monsoon, 2 months SPI for November. In this
manner, the occurrence of drought based on the SPI and SMDI in different time intervals  was  discussed together.

Overall, the soil moisture conditions can be influenced by irrigation in plain areas of Terai. We have not considered irrigation
and artificial water storage while setting up the model. In those areas, the supplemental irrigation might have elevated the soil
moisture level in irrigated fields. Similarly, the soil moisture derived from the model was not validated independently due to
the lack of the observed data and validation was only limited to discharge and evaporation data.4 Results

**4.1 Calibration and validation of Hydrological modelling**

**4.1.1 Validation with potential evapotranspiration data**

The PET validation was performed at three observed evaporation stations, at Kathmandu, Okhaldhunga, and Jiri in Nepal,
where the PET was estimated using a class-A pan. These three stations were chosen for the validation of PET as they have
data for a longer period with little missing data. The stations also depict elevations between 1,300–2,000 m asl. The J2000
model calculates daily potential evapotranspiration using the Penman-Monteith equation (Allen et al., 1998). Further, these
daily values were aggregated to monthly sums and compared with the observed data.
The graphical plots (time series and scatter plots), as well as the coefficient of determination ($R^2$), show that the model has
simulated monthly PET at Okhaldhunga and Jiri stations relatively better than at the Kathmandu station (Figure 3, Table 1).
Although the monthly variability is captured well in Kathmandu, the over-prediction in springtime is evident. This might be
related to the fact that about 25% of the data was missing, a higher proportion as compared to the other two stations (15% in
Jiri and 3% in Okhaldhunga). The overall amount of PET is captured well by the model indicated by PBIAS derivations from
–0.05% to 9.4%.

### 4.1.2 Validation with discharge data

The calibration was carried out using daily discharge data from 1985 to 1995 and validation was carried out from 1996 to
2007 using measurements at the Chatara discharge station. The calibrated parameters of the J2000 model for the Koshi river
and their range are listed in Supplementary Table 3.

Figure 4 shows the comparison between observed and simulated daily streamflow at Chatara for the calibration and
validation periods. Table 2 shows the statistical evaluation based on four chosen objective functions. Figure 4 indicates that
the model reproduced the overall trend of observed data in the calibration period, which has been reflected in the NSE
(0.95), KGE (0.93), and $R^2$ (0.95). However, there is some underestimation, especially during the flood season, during most
of the initial years. The PBIAS is –4.6% during the calibration period, indicating reasonable model simulation with slight
underestimation. During the validation period, the overall hydrograph pattern is represented well as indicated by the NSE
(0.91), KGE (0.91), and $R^2$ (0.92). However, the discharge is underestimated in 2002 and 2003 and overestimated in 2001.
The simulated flow is overestimated (PBIAS = 6.5%) during the validation period. The slightly lower model performance in
the validation period is indeed expected due to variations in meteorological variables (for example, rainfall) during the
calibration and validation periods. Overall, the model has represented patterns of base flow conditions and rising and
recession limbs during both the calibration and validation periods.

## 4.2 Spatial and temporal variability of the soil moisture conditions

The temporal and spatial variability of soil moisture are mainly influenced by two kinds of factors. Precipitation, on the supply
side, increases soil moisture. On the demand side, land use and land cover, temperature, and other climatic variables decrease
the moisture content of the soil. Higher temperatures could increase evaporation and transpiration from the soil. Here, we
discuss the temporal and spatial variability of precipitation and temperature of the river basin, soil moisture variability, and
the soil moisture drought index, as simulated by the calibrated and validated model.

### 4.2.1 Temporal variability of precipitation and temperature

The SMDI is calculated for the three study regions in the KRB (Figure 1) — trans-Himalaya, mountains and plains. Most of
the variation in the soil moisture is due to the dynamic relationship of precipitation and temperature and other variables within
the basin. Figure 5 shows the seasonal precipitation for the three regions. The precipitation in trans-Himalaya region is very
low in all the seasons compared to the other two regions. On average, the precipitation in trans-Himalaya region is about 27%
of that in the high and middle mountains and 35% of that in the plains. The variation in precipitation is the highest for the
plains, especially during the post-monsoon season. The mountains and the plains get the highest precipitation during the

monsoon season. The three lowest years of precipitation are highlighted in red bars for all the seasons. There was no rainfall in the winters of 1999 and 2006 in the plains.

Figure 6 shows the seasonal temperature anomalies for the three regions. The long-term average seasonal temperature for the regions is marked by the numeric value coloured in red. The trans-Himalaya exhibits the lowest temperature for all the seasons while the plains shows the highest temperature for all the seasons with the mountains in between. There is a steady increase in the average temperature throughout the basin for all seasons. The average temperature shows a positive anomaly after 2000 in the winter and monsoon seasons in the trans-Himalaya. A similar rise in average temperature can also be seen from 1993 in

the monsoon and 1998 in the winter season in the mountains. The pre-monsoon average temperature is also above average in the plains, after 1999.

### 4.2.2 Spatio -temporal variability of soil moisture

Supplementary Figure 2 shows the variation in weekly soil moisture for the KRB. Supplementary Figure 2 show the variation in soil moisture in each week; the most stressful period is around the pre-monsoon season. This is mainly due to low rainfall

and high temperatures at that time of year, which causes higher evapotranspiration and less soil moisture. When precipitation begins during the monsoon season, the soil water content increases and saturation is reached at the maximum level. After the post-monsoon season, the soil moisture starts decreasing until the pre-monsoon season of the following year. About 3% of the basin's area is glaciated, and not considered for the analysis of soil moisture as there is no interaction between the glacier module and the soil module in the model.

Figure 7 shows the spatial average weekly variability of the SMDI for the three regions from 1980 to 2007. Severe drought conditions (having SMDI values below –2.0) during the whole period are marked in red dots. Such values are more frequent in trans-Himalaya and the plains. In the plains, moderate drought events have increased in recent years, especially after 1998. To demonstrate spatial and temporal variability, Table 3 shows the average SMDI values for the three physiographic regions and the whole basin from 1980 to 2007. At the basin scale, the year 1992 was the driest (SMDI: -0.91) followed by 1994 and

2006. Similarly, the wettest year was 1998 (SMDI: +0.82), followed by 1980 and 1996. However, for each physiographic region, the dry and wet years are different and are discussed in the sections below specific to the physiographic region. Similarly, Figure 8 shows the spatial maps of average annual SMDI of the whole Koshi river basin from 1980-2007, including the dryest and wettest year. The average SMDI value normal range from -1.0 to 1.0 for the whole basin. The average value of the trans-Himalaya is normally 0 to -1.0 while mountains and plains show more wetness. However, table 3 shows the great

variability of SMDI in which 16 years shows negative SMDI and 12 shows positive. Compared to the average annual SMDI values, the year 1992 shows that most of the areas in dry conditions as shown in the figures (right upper panel). There are certain areas where the average SMDI is below 2.0 suggesting severe to extremely dry conditions. As given in Table 3, on

average, 1992  was the driest year (SMDI: -0.91) among the 28 years whereas year 1998 was the wettest year (SMDI: 0.82) (1998, right lower panel).

To demonstrate spatial and temporal variability, the variation in SMDI is discussed for three physiographic regions and four seasons: winter, pre-monsoon, monsoon, and post-monsoon.

*Spatial and temporal variability of SMDI in the trans-Himalaya*

The interannual variability of the SMDI in the trans-Himalaya region for all the four seasons is shown in Figure 9 ( top). Dry conditions (SMDI value below –1) are highly prominent during the winter and pre-monsoon seasons, and to a lesser degree during the post-monsoon season. The trans-Himalaya region is in a dry condition for most of the year especially during 1983–1995, and 2001–2007 during the winter and pre-monsoon seasons. More than half the total area of the trans-Himalaya region is under dry conditions between 2001 and 2007 in winter and in the pre-monsoon season during 1989–1992, 1994-1995, and

2001–2007. During the monsoon season, dry conditions are prevalent throughout the study period except for 1980, 1981, 1996, and 1998–2000. The occurrence of dry conditions is erratic in the post-monsoon season but with high spatial coverage, more than 70% in 1982, 1991, and 1994, and about 50% for 2001 to 2006. At the annual level, 21 years showed negative SMDI values and 7 showed positive. The year 2006 was the driest followed by 2002 and 1983. Similarly, the year 1980 was the wettest followed by 2000 and 1999. The consecutive dry years are prominent in trans-Himalaya which can be seen by in three

occasions: 1982-1987, 1989-1995 and 2001-2007.

*Spatial and temporal variability of SMDI in the mountains*

The interannual variability of the SMDI in the mountains for all four seasons is shown in Figure 9 (middle). Dry conditions

(below an SMDI value of –1) are prominent in the pre-monsoon and post-monsoon seasons. The winter season shows wet conditions (above an SMDI value of 1) for most years except during 1980, 1981, 1990, 2000, and 2005. However, about 50% of the total area of the region experienced dry conditions in those years as well. More than 50% of the area experienced dry conditions during the pre-monsoon season in 1980, 1988, 1991, 1992, 1995, and 1999.

The monsoon season is largely wet except for the years 1992 and 2005 when about 50% of the area observed dry conditions.

The post-monsoon season shows high variability regarding dry and wet conditions in this region with dry conditions prevalent in 1981, 1984, 1988, 1991, 1994, and 2000. The area under dry conditions seems to go up to 75% in some of these years. The mountains receive the highest amount of precipitation in the KRB (Figure 2; Supplementary Table 2), which results in a higher amount of soil moisture in the region.

At the annual level, 11 years showed negative and 17 showed positive SMDI values. The year 1992 was the driest year followed by 1991. Similarly, the year 1986 was the wettest year followed by 1998. The magnitude of annual SMDI values has not crossed below –1.0 and above 1.0 in any years.

*Spatial and temporal variability of SMDI in the plains*

The interannual variability of the SMDI in the plains for all four seasons is shown in Figure 9 ( bottom). Dry conditions (below an SMDI value of –1) dominate in the pre-monsoon and post-monsoon seasons. During the winter season, wet conditions (above an SMDI value of +1) prevail for most years except 1982, 1992, 1994, 2005, and 2007. Between 30%–50% of the total area of the region is under dry conditions in those years in the winter. In the pre-monsoon season, most of the area is in severely dry conditions (below an SMDI value of –2) from 1991 to 1996. The monsoon season is largely wet in this region except during 1992, 1994, and 2005, when about 80% of the area has dry conditions, with some of the area under extreme dry condition (below an SMDI value of –3) in 1998.

The post-monsoon season shows high variability in soil moisture conditions in this region with dry conditions prevalent in 1981, 1984, 1988–1994, 1997, 2000, and 2004–2007. The area under severe dry conditions seems to have increased to 50% in 1988 and 1994. The plains region receives a fair amount of rainfall with high variability in the volume of rainfall in all the seasons. The temperature is also the highest in the plains in all seasons (Figure 2; Supplementary Table 2). No precipitation is recorded in winter in 1998, 2005, and 2007 (Figure 2), which directly translates into severe dry conditions during those years in the region. Winter temperatures also show positive anomalies after 1997 (Figure 6), except during 2002.

At the annual level, 13 years showed negative SMDI values and 15 showed positive. The year 1995 was the driest year followed by 1994 and 1992. Similarly, the year 2003 was the wettest year followed by 1998 and 1987. A consecutive dryness occurred from 1991-1997

### 4.2.3 Comparison of SMDI and SPI

Figure 10 shows the SPI values for the four seasons and three regions during 1980–2007. The positive SPI values indicate a prevalence of higher precipitation than the long-term average and negative values indicate lower precipitation than the long-term average. Comparing SPI figures with SMDI (Figure 9) indicates that SMDI shows a higher variation of soil moisture conditions than SPI for the same period.

In the trans-Himalaya, the period after 2001 has positive SPI values (Figure 10, top) in the pre-monsoon season in most areas whereas the SMDI (Figure 9, top) shows moderate to extreme dry conditions. In the winter season, the SMDI shows a higher degree of dryness than the SPI. In 1999 (winter), although the SPI is very low (one of the three lowest precipitation years), the SMDI shows wetness in much of the area. Although 2006 (winter) shows the lowest SPI, only 25% of the area is under the severe dry conditions as per the SMDI value. Only in some years or in seasons therein do both the SPI and SMDI indicate similar dry conditions, such as in the winter of 2006, the pre-monsoon season of 1984, 1994, and 1996, the monsoon season

in 1982, 1983, 1994, 2005, and 2006, and the post-monsoon season in 1981, 1991, and 1994. Figure 5 also indicates one of the lowest levels of precipitation during these periods.

In the mountains, the SMDI (Figure 9, middle) shows a higher variation in soil moisture conditions as compared to the SPI (Figure 10, middle). In 1999 (winter), the SPI shows extreme values (below –2) in 50% of the area but the SMDI shows
moderate to severe values in the equivalent area. It is only in some years that both the SPI and the SMDI indicate matching dry conditions—2006 (winter); 1992, 1995, 1996, and 2005 (pre-monsoon); 1992 and 2005 (monsoon); and 1981, 1984, 1991, and 1994 (post-monsoon). These periods also have the lowest rainfall as indicated in Figure 5.

In the plains as well, the SMDI (Figure 9,  bottom) shows a higher variation in soil moisture conditions compared to the SPI (Figure 10,  bottom). In the pre-monsoon and post-monsoon seasons, after 2004, the SPI shows normal conditions in the
majority of the areas, whereas SMDI shows moderate dry conditions. During 1996–2004 (monsoon), the SPI shows normal conditions whereas the SMDI shows moderate wet conditions. Only in some years, both the SPI and SMDI indicate matching soil moisture conditions: 2001 and 2006 (winter); 1994, 1995, and 1996 (pre-monsoon); 1992, 1994, and 2005 (monsoon); and 1981, 1984, 1991, and 1994 (post-monsoon). These periods also have the lowest rainfall as indicated in Figure 5.

### 4.2.4    Magnitude, duration and extent of drought events

SMDI values lower than –3.0 are considered here as extreme soil moisture-deficit conditions and can be interpreted as a 'drought'. To understand the temporal extent of droughts in the KRB, weekly events as a percentage of total weeks in a given season when the three regions are under extreme drought  This shows the temporal variation in the duration of drought events. The total number of weeks for each season are: winter and pre-monsoon (13 weeks each), monsoon (18 weeks) and post-monsoon (8 weeks).

Figure 11 shows the spatial maps of the duration of severe droughts in a number of weeks, for annual average, driest and wettest year. In average, drought prevails for 1-5 weeks in the mountains and plains, and in some patches up to 10 weeks mostly in the plains. In trans-Himalaya, drought prevails up to 10 weeks. However, the driest year 1992 has drought up to 36 weeks mostly in the western part of the trans-Himalaya and some patches on mountains and plains. The wettest year, 1998 mostly shows drought weeks limited to 5 weeks, although central part of the Koshi in plains shows drought weeks up to 36.
This shows that even during the wettest period, some localised drought events may prevail.

Table 4 shows the duration of drought events in a number of weeks when SMDI value is below -3.0. The table indicates that year 1992 has the maximum number of weeks (i.e. 8.5) of drought events followed by 1995 and 2006 at the basin scale. The duration of drought events is different in physiographic regions. For trans-Himalaya, the maximum drought events occurred in 2006 followed by 1983 and 1992. For mountains, it is 1992, followed by 1991 and for plains, it is 1995 followed by 1994
and 1992. Among the three physiographic regions, the trans-Himalaya has the higher frequency of average duration of drought events (4.9 weeks) followed by plains (3.6 weeks) and mountain (2.9 weeks) from 1980-2007.

Figure 12 shows the percentage of weeks with drought in the trans-Himalaya ( top), mountains (middle), and plains ( bottom). In the trans-Himalaya, droughts are prominent in the pre-monsoon and winter seasons. A continuous drought can be seen during 2001–2007. In particular, in 50% of the area, a drought occurred in at least half the pre-monsoon period (and up to 90% of the area in some places). In 2002 and 2007, more than 70% of winter weeks are under drought. Pre-monsoon drought is also frequent in all the years except 1980–1982 and 1996–2000. In the monsoon season, about 25% of the weeks witness drought in most years, with a few exceptions. Frequent droughts are also observed in 1982, 1991, 1994, and 2006 in the post-monsoon season. In 1982, more than 60% of the area has a drought in about 90% of the weeks.

In the mountains (Figure 12, middle), drought is most prominent in the pre-monsoon and winter seasons. Continuous drought can be seen for about one-third of the winter season over about 15% of the land area every year, and in some years up to 25%-40%. Severe droughts are seen more frequently in the pre-monsoon season and over a wider area. In some years, such as in 1989, 1992, 1995, and 2006, a drought occurred over more than 50% of the area and up to 75% in 1992. In the monsoon season, a smaller area is under drought as this region receives the highest precipitation then (Figure 2; Supplementary Table 2). Droughts are less severe in the post-monsoon season, as compared to the pre-monsoon season. However, there are cases of drought in 40% of the weeks in 1991 and 1994. In 1991, this was in 25% of the region and up to 50% in 1994.

In the southern plains (Figure 12,  bottom), drought is prominent in the pre-monsoon and winter seasons. The magnitude of drought is higher in the pre-monsoon season than the winter season. There are continuous drought events from 1989 to 1997 where, in 40% of the pre-monsoon weeks, the drought extends to more than 50% of the area, and in some years, up to 75% of the area. In 1995 in particular, up to half of the region's area has drought for about 90%–100% of the pre-monsoon period. The drought is only visible in about 10% of the monsoon period in about 25% of the area. In the post-monsoon season of 1988 and 1994, nearly 75% of this region experiences drought for more than 40% of the weeks. This higher incidence of drought in the plains is mainly due to it having the highest temperatures among the three regions of the KRB (Figure 2; Supplementary Table 2).

We also looked at the extent of maximum and average drought coverage for different seasons Figure 13. Here, we calculated the maximum area covered by drought in any particular week of the season and the average area overall weeks of the season. Figure 13 shows the average area (black line) and maximum area (red line) covered by drought as a percentage of the total area.

While the variability of maximum area coverage in the trans-Himalaya region and the plains is higher than in the mountain regions, the pre-monsoon season in the mountains also shows a higher degree of variability compared to other seasons. Although the average area affected by drought is lower during the monsoon season in all the regions, the maximum area coverage is higher than other seasons and, in some years, have reached more than 50% of the area in the trans-Himalaya and the plains. This indicates that, although wetness prevails in the monsoon season, drought could reach more than 25% of the region's area for at least one week. During the post-monsoon season and in winter, the average area and maximum areal

coverage have smaller differences, which indicates that the spatial coverage of drought prevails in most of the region during
these seasons.

## 5 Discussion

The application of the J2000 model in transboundary  Koshi river basin with the three physiographic regions has enabled to understand the spatial and temporal variation of soil moisture conditions and related droughts. The distinct pattern of soil
moisture influenced by both temperature and precipitation conditions are reflected in four seasons and distinct physiographic conditions.

In the trans-Himalaya region, the dry conditions in the winter season from 2001–2006 may be attributed to the low winter precipitation (Figure 5). Three of the lowest precipitation years during the study period occurred after 1998 (1998, 2005, and 2007). The average surface temperature has also steadily increased in the winter season during this period. Only positive
temperature anomalies are observed after 1998 in the winter season. In the pre-monsoon season, the dry conditions are probably derived from the temperature, which increased after 1998 up to 2004 (Figure 6). The three lowest years of monsoon precipitation occurred during 1982, 1983, and 2006, which coincides with the dry conditions in that period. A positive temperature anomaly is seen during the monsoon after 1987 barring a few years such as 1992, 1996, and 1999, which also translates into dry conditions during those periods. However, the interannual variation in precipitation is low for the monsoon
season in the region. The data shows a positive post-monsoon temperature anomaly after 1999, except for 2004, which translates into the dry conditions in that period. Post-monsoon precipitation is highly variable in the region leading to high interannual variability in dryness in the region.

In the mountains, the three years with the lowest precipitation in the winter season were 1998, 2004, and 2007 (Figure 5), which directly translates into dry conditions in the region. The winter temperature also shows positive anomalies after 1997
(Figure 6). The three years with the lowest precipitation in the pre-monsoon season were 1992, 1995, and 1996, whereas positive temperature anomalies can be seen for most years after 1990. This correlates with the dry conditions in those periods in the region. Post-monsoon precipitation is highly variable in this region (Figure 5). The three years with the lowest post-monsoon precipitation are 1981, 1991, and 1994. The temperature anomalies are also positive during 1998–2003, which is one of the reasons for the dry conditions in this period.

In the plains, no precipitation was recorded in winter of 1998, 2005, and 2007 (Figure 2), which directly translates into severe dry conditions during those years in the region. Winter temperatures also show positive anomalies after 1997 (Figure 6), except during 2002. The three years with the lowest precipitation in the pre-monsoon season, and a consequent positive temperature anomaly, were 1994–1996. This correlates with dry conditions in those periods. Only positive temperature anomalies can be seen in the pre-monsoon season after 1998. The dry conditions in the post-monsoon season may be attributed to the highly

variable precipitation in this region (Figure 5) with values ranging between 50–300 mm. The three years with the lowest post-monsoon precipitation were 1981, 1984, and 1997. The temperature anomalies are also positive in most years after 1992. From the period of 1981-2007, the year 1992 is the driest year over the whole basin and the maximum number of weeks of drought occurrence (i.e 8.5 weeks). On contrary, the year 1998 is the wettest year and lowest number of weeks of drought occurrence (i.e  1.2 weeks (Figure 8 and 11; and Table 3 and 4).

Analysis related to SMDI and SPI, the former is able to reflect variations in soil moisture conditions better than SPI which shows normal conditions. As shown in trans-Himalaya, the period after 2001 when SPI shows wetness and SMDI show dryness during the pre-monsoon.  It is because SMDI incorporates additional variables (temperature, evaporation, vegetation, root depth, and soil water holding capacities) to calculate soil moisture variability compared to only precipitation variables by SPI. As expected, the SPI gives a more homogeneous response because of the lack of the representation of physiographic

differences. An example of this behaviour can be seen in winter 2006 where SPI indicates a severe drought in over 80 % of the area of trans-Himalaya and mountains (Figure 9). In contrast, SMDI shows a more differentiated pattern (Figure 8) where during winter drought conditions are indicated for roughly half of the area with severe values only for 10 to 20% of the area. Most likely, one reason for this more differentiated picture is the consideration of soil water storage in the SMDI. The remaining soil water after the post-monsoon can be very important for the water supply and overshadow the effect of (missing)

precipitation in winter. Additionally, this effect can be amplified by the low ET volumes during winter (Figure 3) that deplete the stored soil moisture only slowly, resulting in higher SMDI values. The shown differences in the SMDI are caused by varying soil water storage capacities which control the duration of periods during which higher SMDI can be maintained without precipitation.  The years for which both SPI and SMDI show matching drought conditions can be mainly attributed to them being the lowest rainfall periods (Figure 5).

At the basin scale, the higher incidence of soil moisture deficit is in the plains which is mainly due to higher temperatures. In the trans-Himalaya, droughts persist for a higher number of weeks in the seasons mainly due to low precipitation. A higher frequency of drought is observed in the winter and pre-monsoon seasons. The monsoon season is least affected by the drought due to abundant precipitation at this time but even so, about one-quarter of the season is affected (Figure 12). There is an increasing trend in the frequency of drought in recent years during the winter and the pre-monsoon season. Similarly, the extent

of maximum area covered under drought is higher during the monsoon season and in some years have covered more than half of the basin area. This indicates that although precipitation brings wetness during the monsoon season, drought could reach more than 25% of the region for at least one week.

*Uncertainties and limitations*

The model results are subject to several uncertainties and limitation which are briefly described below. The calibration and validation of hydrological model results are subject to uncertainty arising from model input data, parameter and structural uncertainty. In the mountainous region, the representation of the observed station network is sparse and limited which is the

case in the KRB. For the northern part of China and southern Indian side, gridded datasets were used compared to station data in the Southern Himalaya in Nepal part. The application of APHRODITE data in the northern region has limited the study period up to 2007 because of the data available only up to this period. The station data are mostly limited to the lower elevation areas with limited station network in high altitude areas. Both the gridded and observation network have their advantage and disadvantages for modelling applications. Nonetheless, our approach of using both gridded and station data along with discharge data have enabled us to use the modelling period of 28 years which is a relatively a longer period in the case of transboundary KRB.

We could not validate soil moisture result with station data due to lack of soil moisture network in the Koshi basin. Validation with remote sensing product was also not reasonable due to differences in soil moisture depth definitions and spatio-temporal resolutions. However, a comparison with CCI SM remote sensing-based soil moisture (Liu et al., 2011; Liu et al., 2012; Dorigo et al. 2017) suggests that both remote sensing and model shows inter-annual variability in soil moisture in which soil moisture is high during the monsoon season and low in the spring season but the absolute volume difference is high. The differences could be due to the different soil moisture depth in CCI SM (40 cm) and the J2000 model (up to 100 cm). Supplementary Figure 3 shows the comparison between CCI SM and J2000 model soil moisture comparison.

Regarding the parameter uncertainty, the application of the J2000 hydrological model in the previous studies has shown the potential of spatial transferability of model parameters within the sub-catchments of the Koshi basin (Nepal et al. 2017). The generalised likelihood uncertainty estimation (GLUE) analysis of two sub-catchments of the Koshi basin suggests that most of the time the parameter uncertainty can be explained within the ensemble range of multiple simulations, except some flood events. Supplementary Table 1 shows the J2000 model parameters including the selected parameters which were used for sensitivity and uncertainty analysis by Nepal et al. (2017). Similarly, there were good matches on the category of high, moderate and low sensitive parameters between the two catchments suggesting the robustness of the model in the Himalayan catchments. The results have also suggested that spatial transferability of model parameters in the neighbouring catchment with similar climatic and hydrological conditions are possible in the Himalayan region (Nepal et al, 2017), however, some variation is parameters can be expected if the scale of the basin size and climatic conditions differ (Eeckman et al. 2019; Shrestha and Nepal, 2019). Besides, the soil moisture from the J2000 model was not validated independently due to the lack of the observed data and validation was only limited to discharge and evaporation data. Despite these uncertainties and limitation, the model has replicated overall hydrological behaviour including both low and high flows, similar to the previous studies in the Koshi basin (Nepal et al. 2014, Nepal et al, 2017; Eeckman et al. 2019).

*Historical incidences of drought*

We also examined historical drought events and their impacts on agriculture based on the published literature. The soil moisture drought derived by our study also matches the historical drought events in Nepal mainly of 2005-2006 (winter) and 1992 and 2005 (summer).

Dahal et al. (2016) and Shrestha et al. (2017) reported dry spells in central Nepal during the winter of 2005–2006 and their implications for agriculture. Our results for the same year also showed that more than 75% of the area in the mountains had an SMDI below –1. Drought (SMDI < –3) occurred in more than half the Koshi River basin's area for more than 40% of the winter. This winter drought of 2005–2006 had the highest spatial coverage in the mountains region over the 28-year period under study (Figures 8 and 10). Dahal et al. (2016) reported less than 30% winter rainfall in 2005–2006, with some areas receiving no precipitation at all. As a consequence, paddy production decreased by 13% compared to the previous year; in some districts in the eastern and central region of Nepal (where the Koshi River basin is located), the reduction in yields was 20%–50%. About 7% of the land under paddy was also reportedly left fallow. Wheat production was adversely affected as well.

As the winter drought of 2005–2006 affected the whole of Nepal, a decrease in paddy and wheat production was also reported from the western region. Subsistence hill and mountain farmers were affected in particular as they tend to be more dependent on rainfed agriculture than farmers in the plains, where irrigation infrastructure is prevalent. Regmi (2007) reported that agricultural production declined by 27%–39% that year in a few districts in the Eastern Development Region compared to the previous year. On average, yields in the Eastern Development Region were about 10% lower than the previous year and almost 15% of the land under paddy was left fallow.

Dahal et al. (2016) and Shrestha et al. (2017) also discussed the summer drought of 2005 in central Nepal. Our analysis also showed the 2005 monsoon drought as the largest in terms of area; more than 50% of the mountains area experienced drought (SMDI < –3.0) in 25% of the weeks (Figure 10).

Bhandari and Panthi (2014) reported the 1992 drought in the monsoon season in western Nepal. The insufficient and untimely rainfall contributed to reduced soil moisture, resulting in an agricultural drought and consequent crop failures. From our analysis, 1992 is reported to have the highest soil moisture deficit for the pre-monsoon and monsoon seasons, during which nearly 90% of the area in the mountains have SMDI values lesser than –1.0, with a higher degree of dryness in the pre-monsoon season (Figure 8). The drought that year (SMDI < –3.0) was the highest for the pre-monsoon season and second-highest for the monsoon season when about 75% and 45% respectively of the basin's area in the mountains experienced droughts for more than 25% of the weeks. Even during the winter of 1992, 40% of the basin's area suffered drought for 25% of the weeks (and over half the winter season in 25% of the area) (Figure 10). Shrestha et al. (2017) also reported the severe summer drought of 1992, based on SPI indices using both observed and satellite data. Shrestha et al. (2000) showed a good agreement between the deficit rainfall in 1992 in Nepal and the El Nino of 1992 and 1993.

Although Bhandari and Panthi (2014)'s analysis was mostly focused on western Nepal, the monsoon's influence extends throughout Nepal, as it passes from eastern through to western Nepal. In the KRB, 1992 was among the three lowest rainfall years in the pre-monsoon and monsoon season. Our assumption is that a similar drought condition have occurred in the eastern mountain districts of the Koshi as well.

Wu et al. (2019) calculated the crop water shortage index (CWSI) based on MODIS-derived evaporation and potential evaporation data for the KRB from 2000 to 2014. The CWSI is found to be consistently increasing from 2000–2006. Our SMDI-based results also indicate a consistent decrease in SMDI since 2001. Although the CSWI and SMDI cannot be directly compared, they both reflect a lack of soil moisture. The year 2006 was found to be one of the severest drought years in both Wu (2019) and our study. Similarly, Hamal et al. 2020 also indicated frequent occurrences of drought in 1992, 1994, 1996, 2001, 2006 which has caused yield loss in whole Nepal.

We did not find information about reported droughts in trans-Himalaya and southern plains for the study period. While the trans-Himalaya part of the KRB has little agriculture land, the presence of irrigation infrastructure in the southern plains makes the context quite different from the mountains, where agriculture is mainly rainfed.

## 6 Conclusions

This study investigated the Soil Moisture Deficit Index (SMDI) in the transboundary Koshi River basin straddling China, Nepal, and India by applying the process-based J2000 hydrological model. The model was calibrated and validated using multi-site evapotranspiration and discharge data. This study presents the first comprehensive results of the spatial and temporal variability of soil moisture for the KRB.

The application of the model has resulted in the following conclusions:

1) The J2000 model can simulate the various parts of the hydrograph for the entire simulation period. However, flood peaks and overall flooding periods have been simulated with slightly lower accuracy for some years.

2) The temporal variability of soil moisture indicated that the highest stress in soil moisture is during the pre-monsoon season.

3) The most severe drought was observed in 1992 throughout the Koshi River basin followed by 1994. The other prominent drought years in the period under study are 2006 and 2002 in the trans-Himalaya region, 1992 and 1991 in the mountains, and 1995 and 1994 in the plains.

4) The year 1992 also has the highest number of weeks (8.5 weeks) under extreme dry conditions, or drought, as characterised by SMDI values lower than −3.0 followed by 1995 (8 weeks). The frequency of these events has increased in the later years of the study period and are most evident in the pre-monsoon season.

5) In the trans-Himalaya, continuous drought persists in the majority of the seasons after 2000. A similar pattern also exists in other regions in the winter and pre-monsoon seasons.

6) The maximum area under drought increases in the plains in the later years of the study period during the monsoon and post-monsoon seasons, in the mountains in the pre-monsoon season, and in trans-Himalaya during the winter and pre-monsoon season.

7) The soil moisture drought derived by our study also matches the historical drought events reported in the literature, mainly the winter drought in 2005–2006, and the summer droughts in 2005 and 1992.

The results also suggest that the SMDI represents soil moisture conditions better than the SPI, as the latter depends only on precipitation. On the other hand, in the SMDI, both precipitation (as a supply) and evapotranspiration (as a demand) have been duly reflected. Our results suggest that the SMDI can provide a better understanding of soil moisture variation and related droughts, and hence might be useful in the agricultural sector, on which millions depend in this entire region. The insights into the duration, spatial coverage, and severity of drought conditions throughout the basin can further provide valuable inputs

towards improved management of water resources and the planning of agricultural production. Also, the understanding of soil moisture processes from this study and response to climatic variables can be expanded to understand the future climate change impact on soil moisture conditions. The comparison with the historic events shows that the achieved results are plausible. Additionally, most of the data necessary to conduct the presented method are globally available. Therefore, we think that we presented a robust and transferable method to estimate spatial and temporal variability of soil moisture drought.

*Code availability*: The source code for the JAMS-J2000 hydrological model and SMDI and SPI calculation are available at http://jams.uni-jena.de/downloads/.

*Data availability:* The model outcome of both hydrological and soil moisture dataset can be made available upon request. The details of model input data are provided in Supplementary Table 2 and can be accessed freely, except a few stations data which was provided by the Department of Hydrology and Meteorology. These observed data are not allowed to distribute publicly by the department.

*Author contribution*: All authors contributed to analysis, writing, review and editing. SN, SP, NS and SK contributed to the

710 conceptualization of the study. SK implemented the SMDI and SPI modules in JAMS modelling system. SN, SP and NS collected model input data, performed simulations and contributed and wrote the original draft and visualization.

*Acknowledgements*. This study was supported by ICIMOD's Koshi Basin Initiative, which contributes to the Sustainable Development Investment Portfolio and is supported by the Australian Aid Programme. ICIMOD gratefully acknowledges the support of its core donors: the governments of Afghanistan, Australia, Austria, Bangladesh, Bhutan, China, India, Myanmar,

Nepal, Norway, Pakistan, Sweden, and Switzerland. Further, the authors acknowledge the support of the German Aerospace Center (DLR) for co-funding the project (grant number D/943/67249358). We also acknowledge the support from Arun B Shrestha, Faisal Mueen Qamar, and Shahriar Wahid. The views and interpretations in this publication are those of the authors and are not necessarily attributable to their organizations. The authors also thank the anonymous reviewers for their critical comments, which helped to improve the article.

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

**Table 1: Goodness-of-fit statistics of PET simulation results for different stations**

| Station | $R^2$ | PBIAS | Elevation (m) | Period | Missing value (observed) |
|---|---|---|---|---|---|
| Kathmandu | 0.56 | –0.05% | 1,336 | 2001–2005 | 25% |
| Jiri | 0.84 | 9.4% | 2,003 | 1990–1993 | 15% |
| Okhaldhunga | 0.71 | 1.1% | 1,720 | 1985–1988 | 3% |


**Table 2: Goodness-of-fit statistics of discharge simulation results during the calibration and validation periods**

| Indicators | Calibration (1985–1995) | Validation (1996–2007) |
|---|---|---|
| KGE | 0.93 | 0.91 |
| NSE | 0.95 | 0.91 |
| $R^2$ | 0.95 | 0.92 |
| PBIAS | −4.6 | 6.5 |

**Table 3: Average annual SMDI values from 1980-2007 for Trans-Himalaya, Mountains and Plains and for the whole Koshi basin. The red and blue bar shows the negative and positive SMDI values, the average SMDI values for each year are given in the respective rows.**

| Year | Trans-Himalaya | Mountains | Plains | Koshi Basin |
|---|---|---|---|---|
| 1980 | 1.49 | 0.21 | 0.57 | 0.72 |
| 1981 | 1.26 | -0.20 | 0.28 | 0.4 |
| 1982 | -0.66 | -0.23 | -0.32 | -0.39 |
| 1983 | -1.21 | 0.45 | -0.04 | -0.21 |
| 1984 | -0.45 | 0.07 | 0.31 | -0.02 |
| 1985 | -0.39 | 0.15 | 0.51 | 0.09 |
| 1986 | -0.29 | 0.96 | 0.18 | 0.33 |
| 1987 | -0.33 | 0.55 | 0.65 | 0.31 |
| 1988 | 0.35 | 0.09 | 0.11 | 0.18 |
| 1989 | -0.29 | 0.04 | -0.17 | -0.13 |
| 1990 | -0.50 | 0.35 | 0.01 | -0.02 |
| 1991 | -0.49 | -0.57 | -0.42 | -0.5 |
| 1992 | -1.08 | -0.78 | -0.91 | -0.91 |
| 1993 | -0.24 | 0.34 | -0.24 | -0.02 |
| 1994 | -1.11 | -0.31 | -0.97 | -0.76 |
| 1995 | -1.00 | -0.30 | -0.98 | -0.72 |
| 1996 | 1.06 | 0.41 | -0.01 | 0.49 |
| 1997 | 0.12 | 0.00 | -0.17 | -0.02 |
| 1998 | 0.92 | 0.71 | 0.87 | 0.82 |
| 1999 | 1.18 | -0.11 | 0.36 | 0.43 |
| 2000 | 1.29 | 0.00 | 0.22 | 0.47 |
| 2001 | -0.84 | -0.09 | -0.28 | -0.38 |
| 2002 | -1.23 | 0.35 | 0.41 | -0.12 |
| 2003 | -0.79 | 0.25 | 1.01 | 0.16 |
| 2004 | -0.87 | 0.43 | 0.24 | -0.03 |
| 2005 | -1.05 | -0.23 | -0.72 | -0.63 |
| 2006 | -1.35 | -0.26 | -0.41 | -0.64 |
| 2007 | -0.28 | 0.62 | 0.39 | 0.27 |




**Table 4: Duration of drought events for trans-Himalaya, Mountains and Plains and for the whole Koshi basin. The red bars corresponding to values on the rows that show the number of weeks when SMDI < -3.0.**

| Year | Trans-Himalaya | Mountains | Plains | Koshi Basin |
|------|----------------|-----------|--------|-------------|
| 1980 | 0.3 | 3.3 | 2.1 | 2.0 |
| 1981 | 0.4 | 4.3 | 1.4 | 2.2 |
| 1982 | 10.6 | 3.7 | 3.3 | 5.7 |
| 1983 | 11.5 | 2.2 | 5.1 | 6.0 |
| 1984 | 5.0 | 2.6 | 2.8 | 3.4 |
| 1985 | 3.2 | 2.4 | 2.9 | 2.8 |
| 1986 | 4.1 | 0.7 | 1.4 | 2.0 |
| 1987 | 5.7 | 1.1 | 1.8 | 2.7 |
| 1988 | 0.7 | 1.7 | 2.9 | 1.8 |
| 1989 | 1.7 | 3.7 | 4.0 | 3.2 |
| 1990 | 3.1 | 1.3 | 1.3 | 1.8 |
| 1991 | 4.7 | 5.4 | 5.8 | 5.3 |
| 1992 | 11.1 | 7.6 | 7.1 | 8.5 |
| 1993 | 1.4 | 1.5 | 4.6 | 2.4 |
| 1994 | 5.0 | 4.0 | 8.5 | 5.7 |
| 1995 | 7.2 | 5.0 | 12.8 | 8.0 |
| 1996 | 0.9 | 2.2 | 5.7 | 2.9 |
| 1997 | 1.2 | 2.4 | 1.7 | 1.8 |
| 1998 | 1.0 | 1.3 | 1.4 | 1.2 |
| 1999 | 0.3 | 4.9 | 2.5 | 2.8 |
| 2000 | 0.3 | 2.0 | 2.7 | 1.7 |
| 2001 | 6.9 | 3.3 | 3.3 | 4.4 |
| 2002 | 10.4 | 0.9 | 0.9 | 3.8 |
| 2003 | 5.5 | 1.9 | 1.4 | 2.9 |
| 2004 | 6.2 | 1.6 | 1.6 | 3.0 |
| 2005 | 6.8 | 3.8 | 5.6 | 5.3 |
| 2006 | 13.9 | 4.7 | 4.2 | 7.4 |
| 2007 | 7.8 | 1.3 | 1.9 | 3.5 |


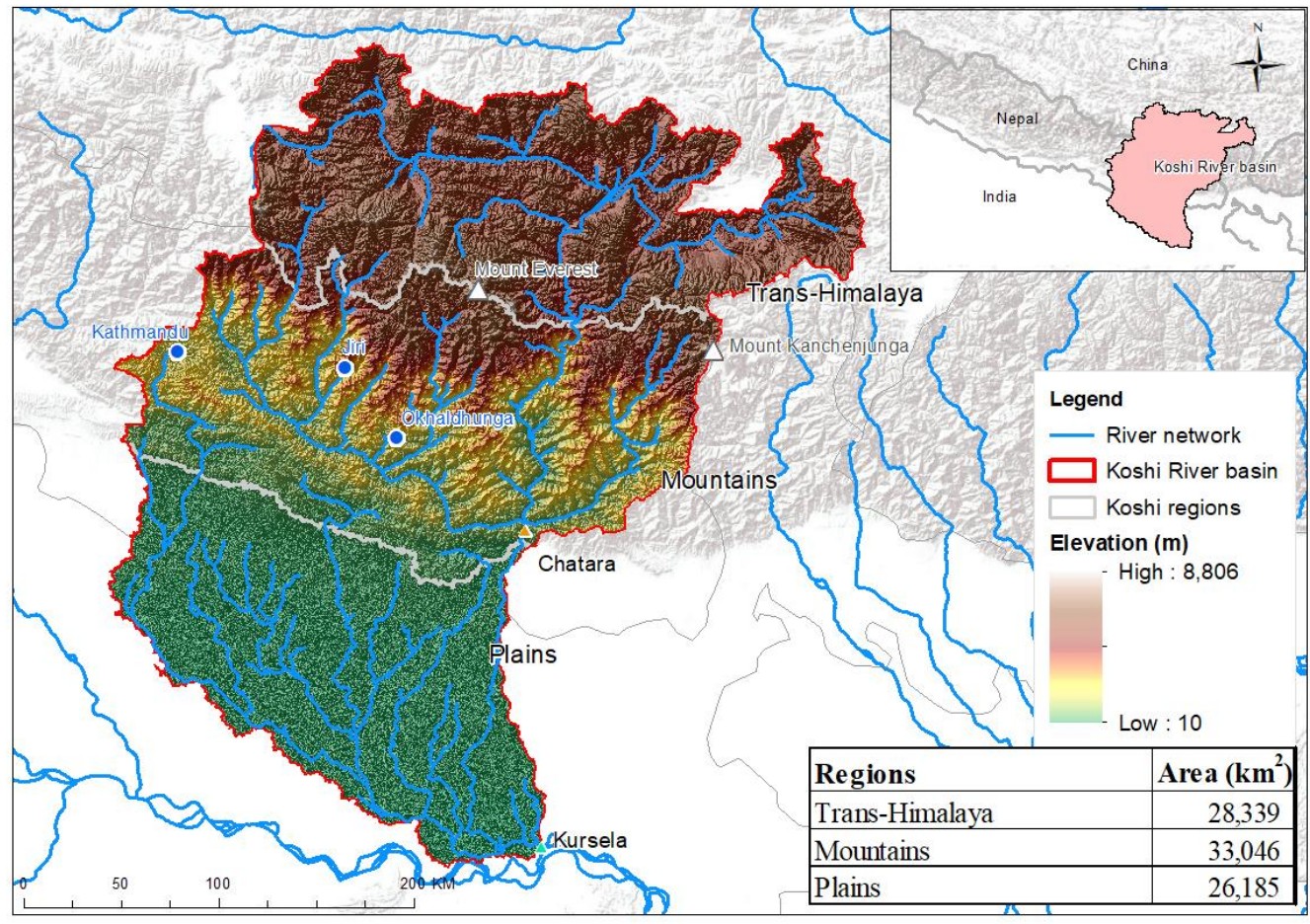

**Figure 1: The transboundary Koshi River basin straddling the trans-Himalaya, the mountains, and the plains** (Note: The map has
used SRTM 90 meter DEM publicly available at http://srtm.csi.cgiar.org/ and ERSI background Terrain map available at ArcMap 10.6.1
software).

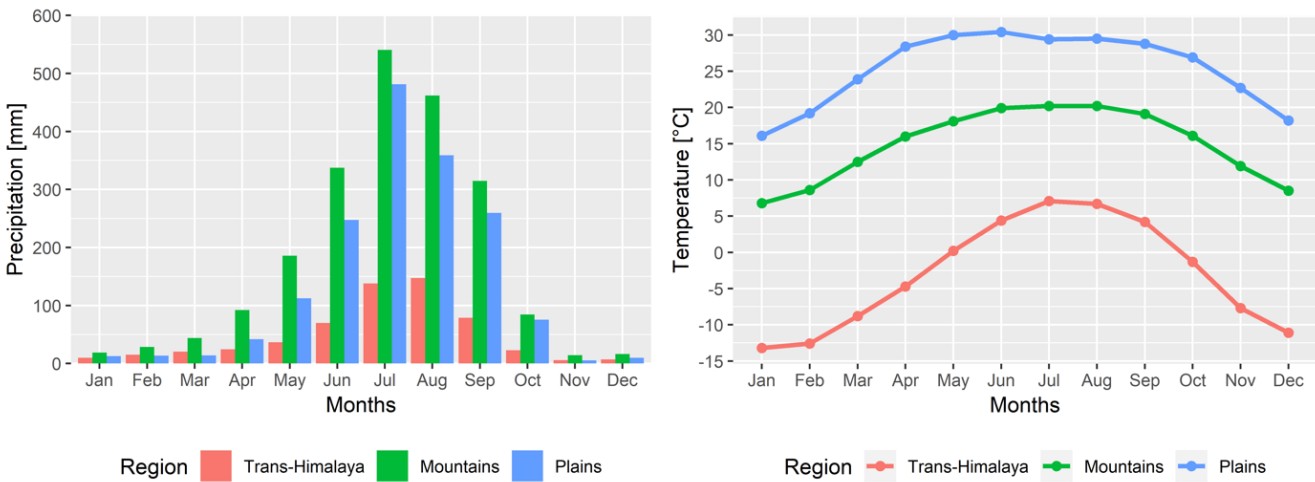

**Figure 2: Monthly precipitation (left) and temperature (right) for the three regions of the Koshi River basin, 1980–2007**


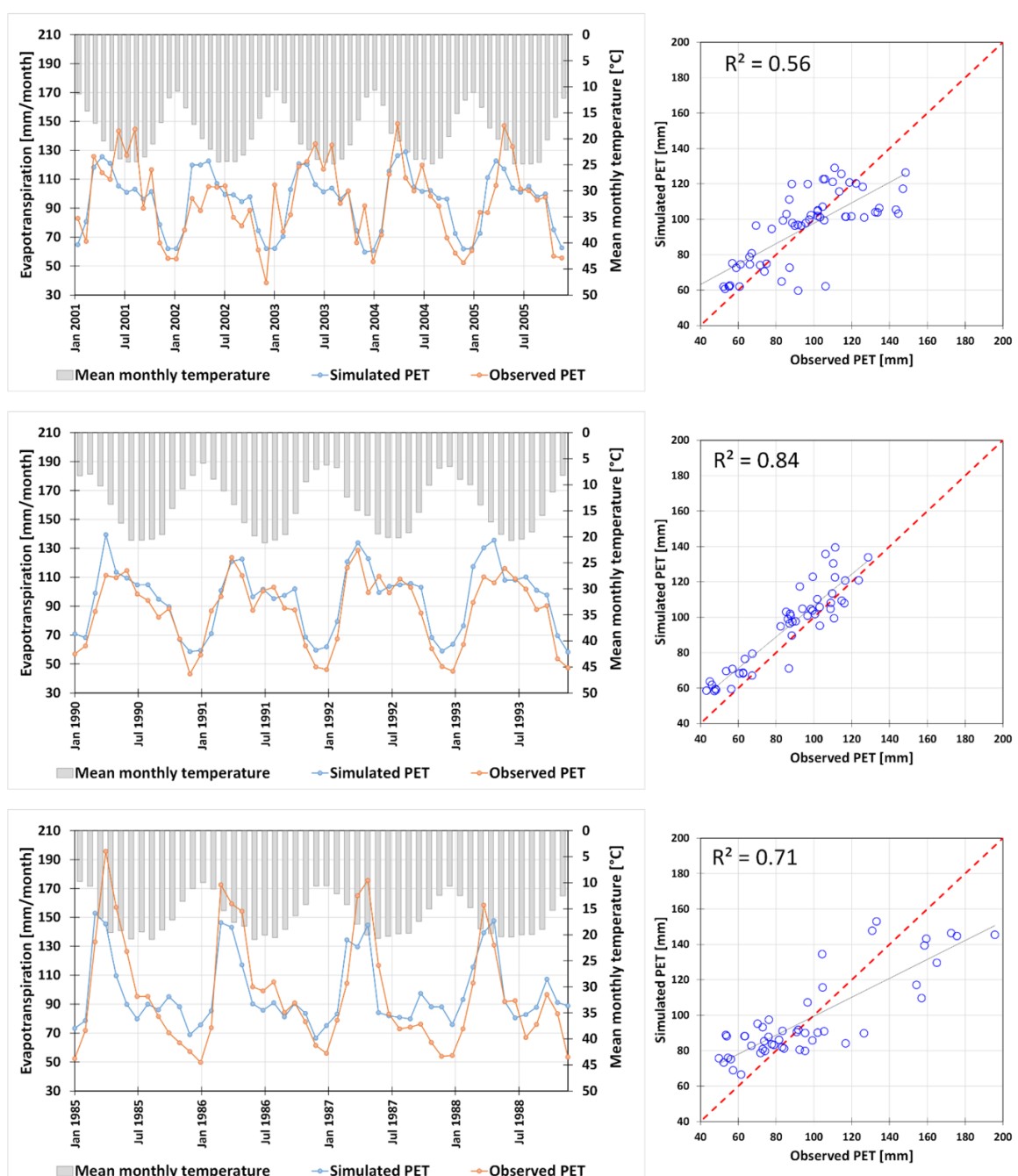

**Figure 3: Comparison of monthly potential evapotranspiration for Kathmandu (top), Jiri (middle), and Okhaldhunga (bottom) stations**

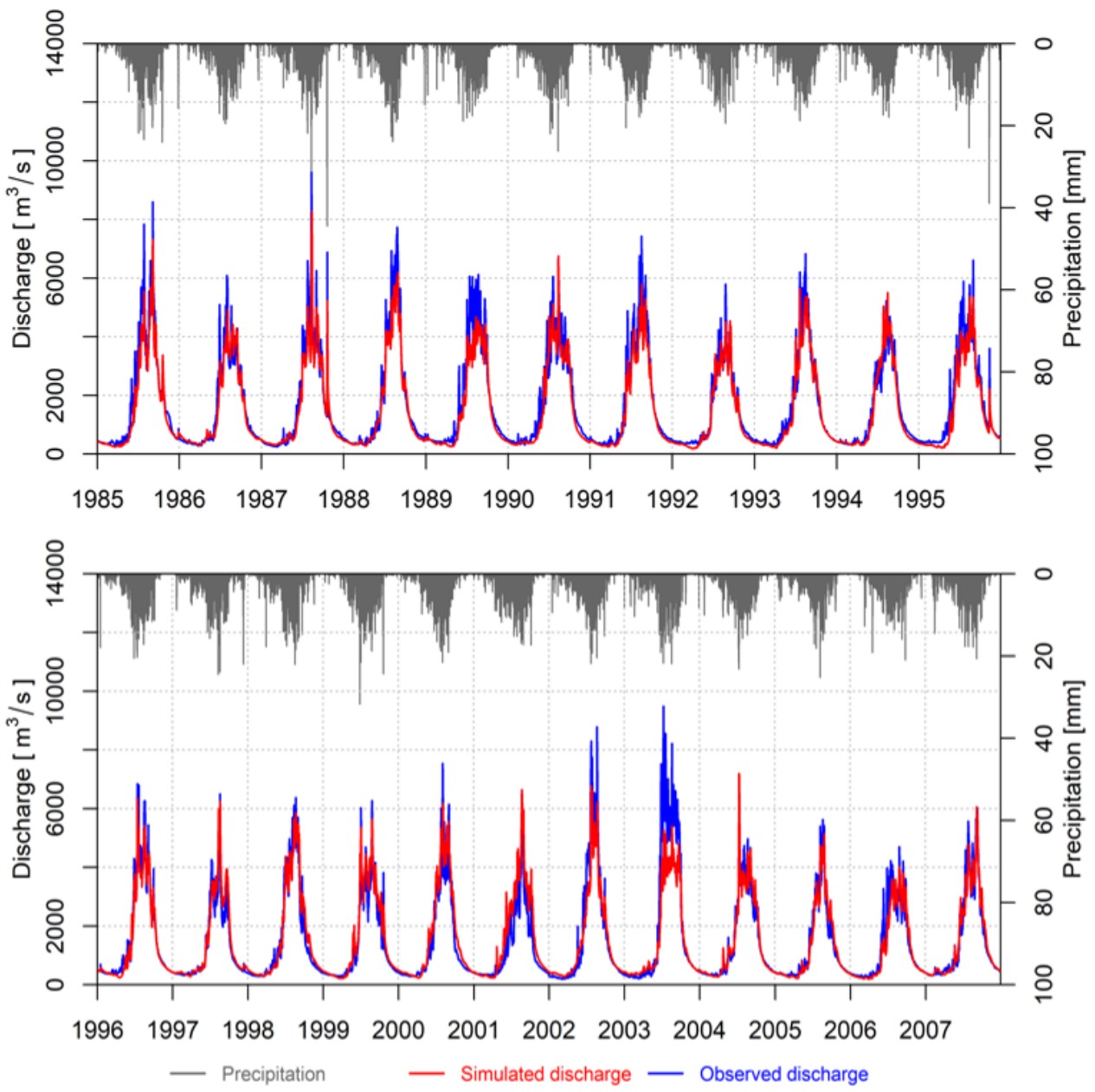

 **Figure 4: Calibration (top) and validation (bottom) of daily streamflow at Chatara**

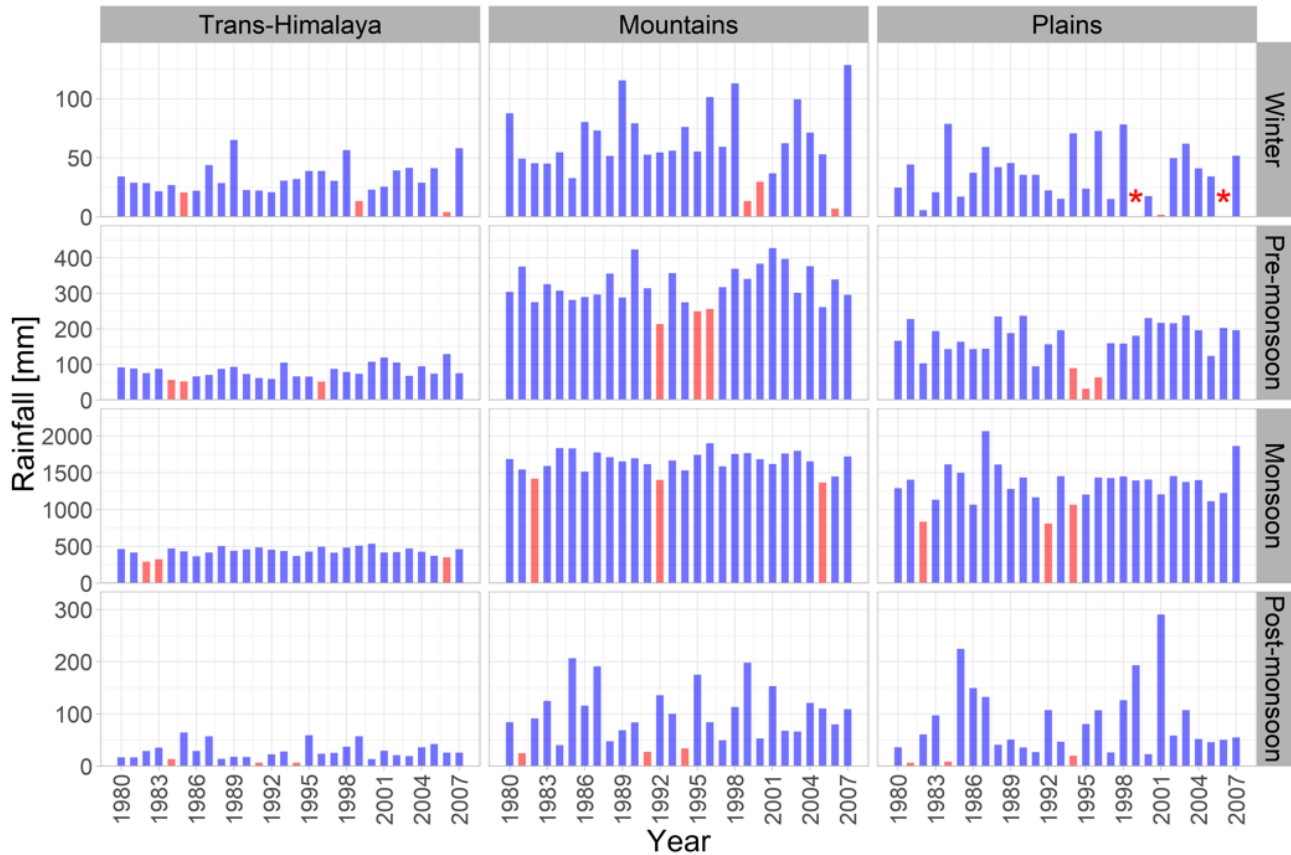

**Figure 5: Seasonal variability in precipitation in the KRB, 1980–2007**

Notes: The red bar indicates the three lowest precipitation years; * indicates no rainfall.


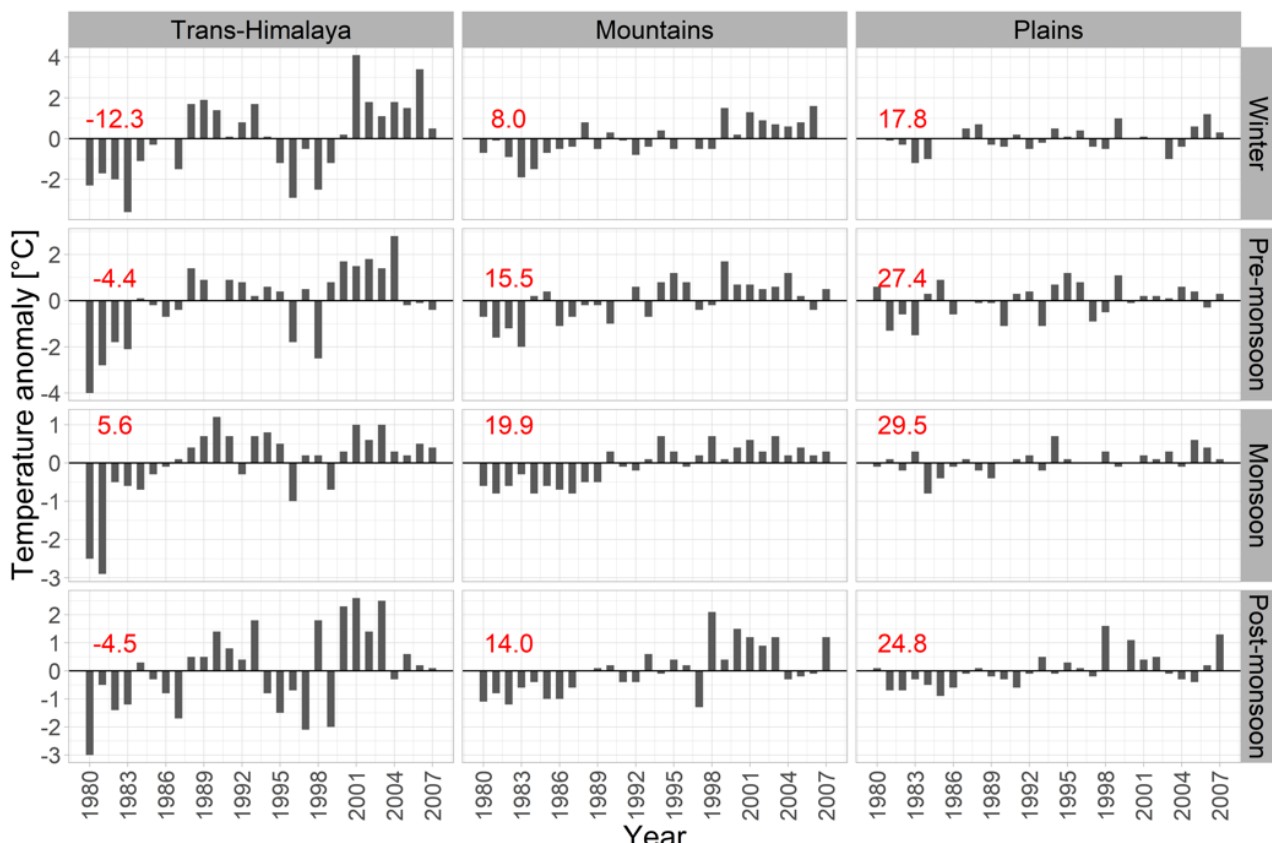

**Figure 6: Seasonal temperature anomalies in the KRB, 1980–2007**

Note: The value in red shows the long-term average annual temperature for each season.


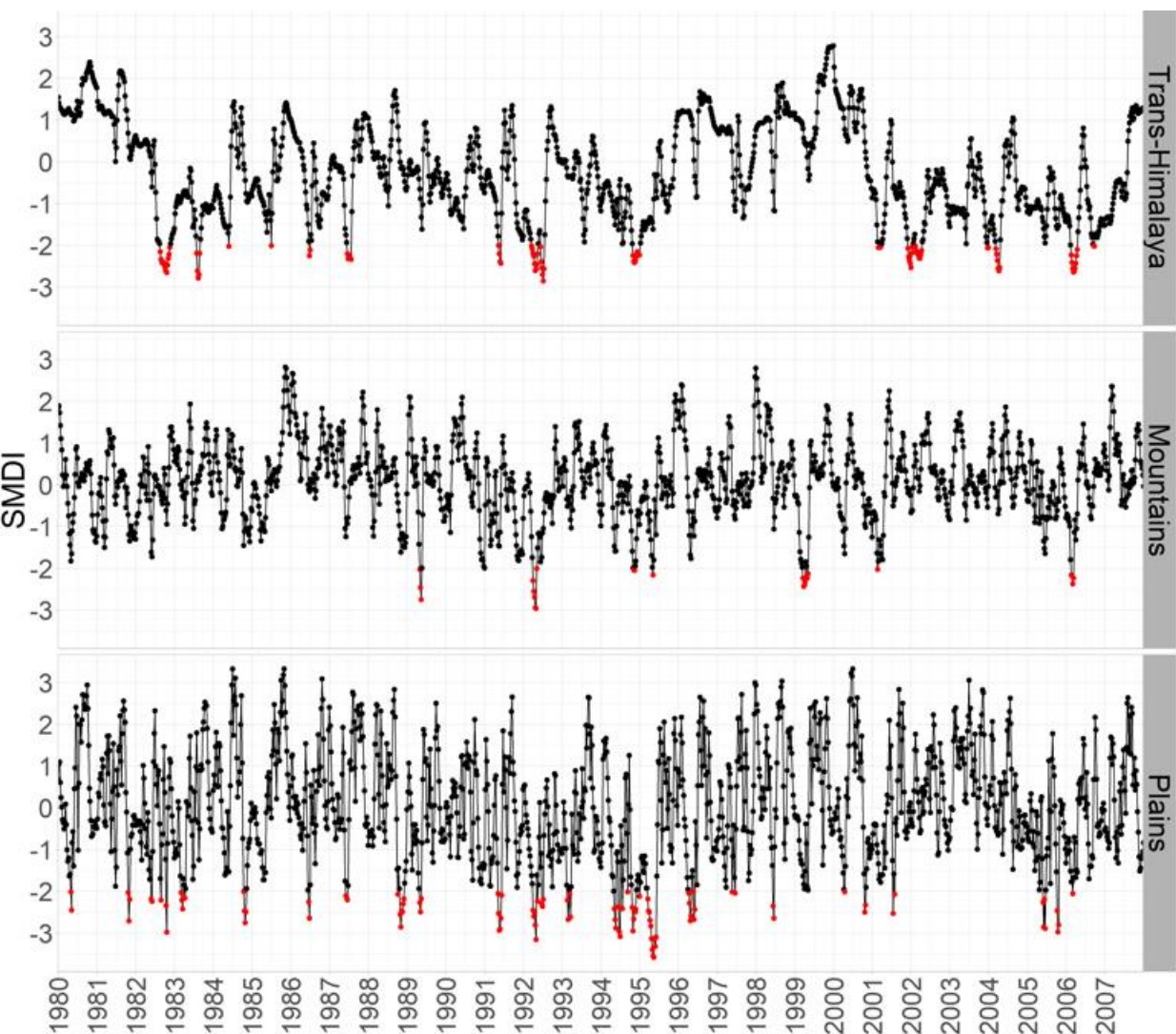

**Figure 7: Average weekly SMDI values for three regions of the KRB**

Note: Red dots show SMDI values below –2.


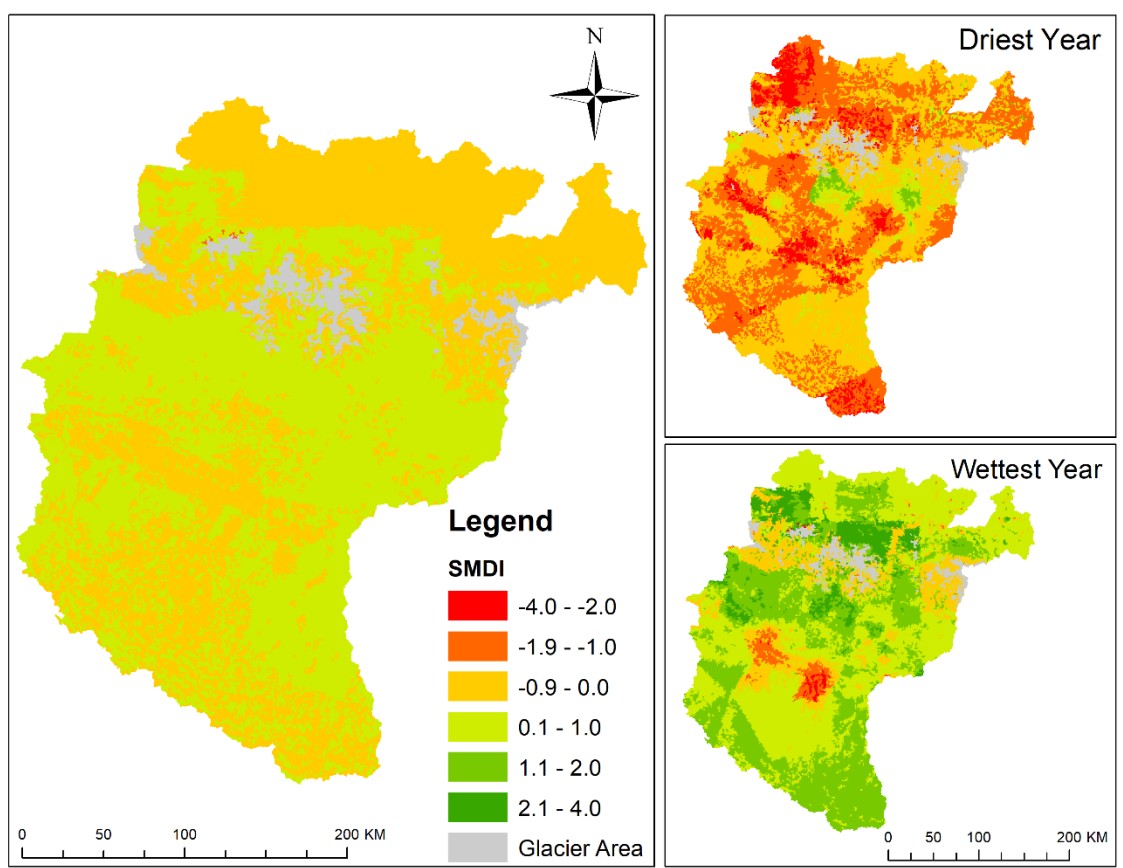

**Figure 8: Spatial maps of average annual SMDI (1980-2007), driest year (1992) and wettest year (1998).**


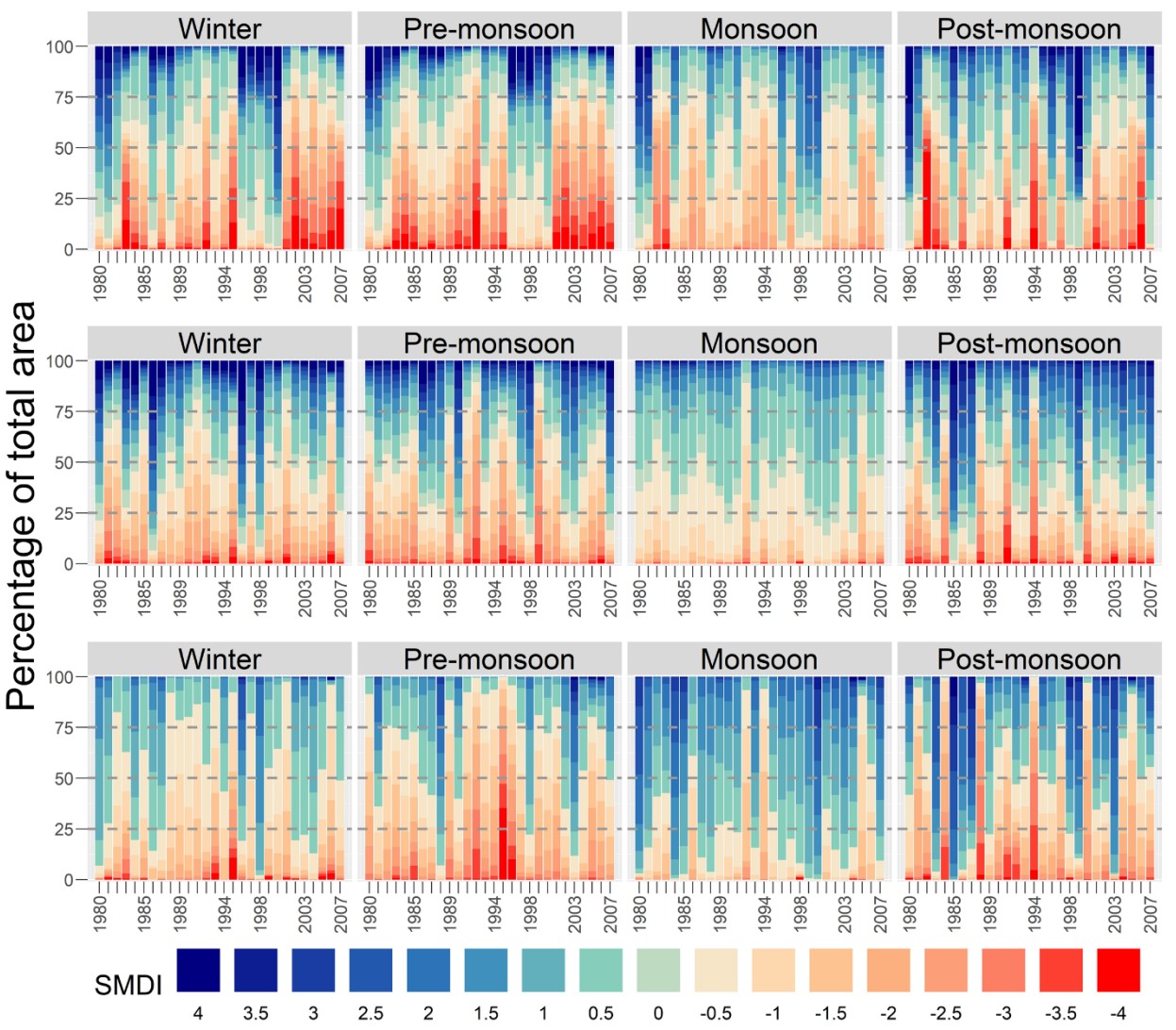

**Figure 9 Spatial and seasonal variability of the SMDI in trans-Himalaya (top), the mountains (middle), and the plains (bottom)**

Note: Each colour band shows the respective HRU's area combined.

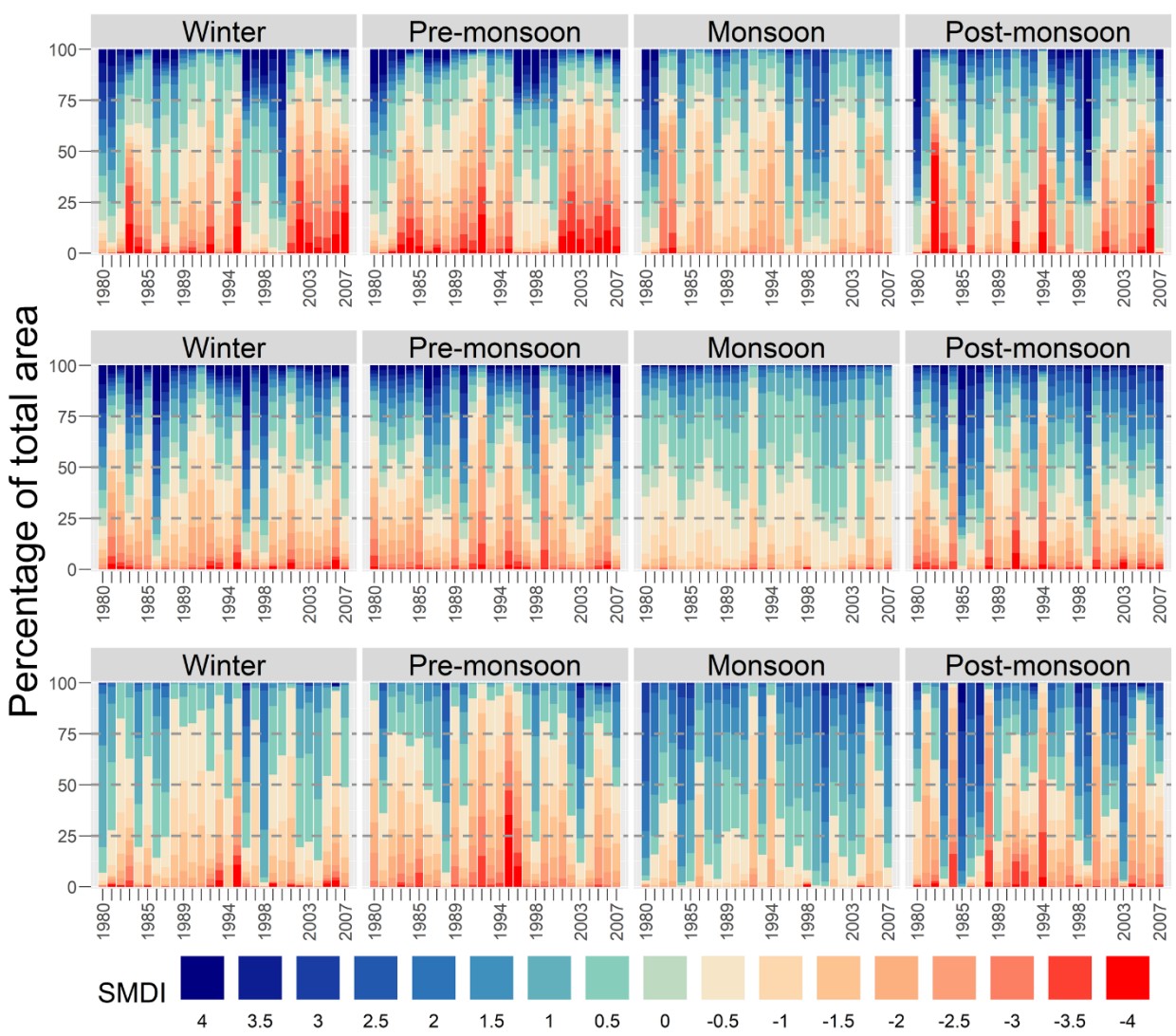

**Figure 10 Spatial and seasonal variability of the SPI in the trans-Himalaya (top), the mountains (middle), and the plains (bottom)**

Note: Each colour band shows the respective HRU's area combined.

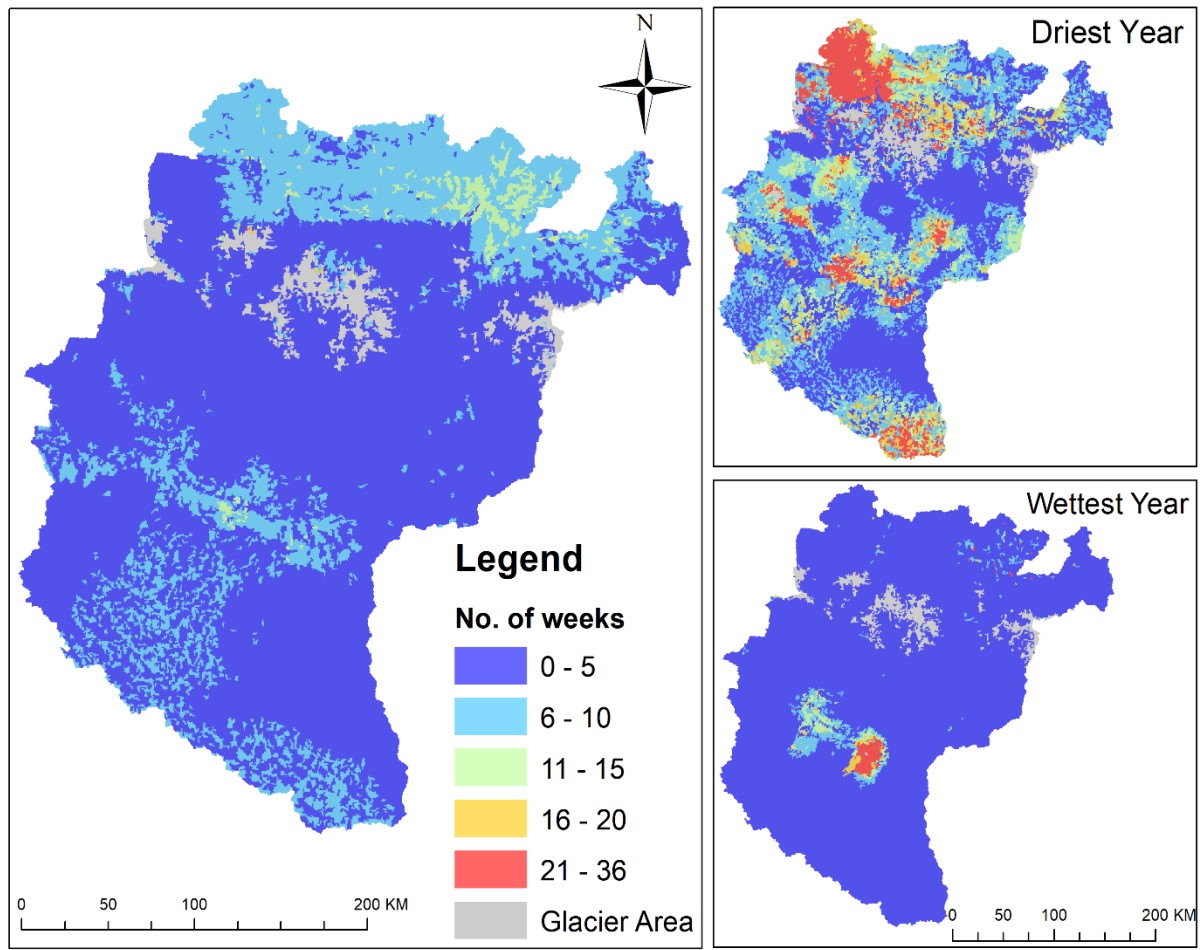

**Figure 11: Spatial maps of average duration of drought i.e. SMDI < -3.0, dryest year (1992) and wettest year (1998).**

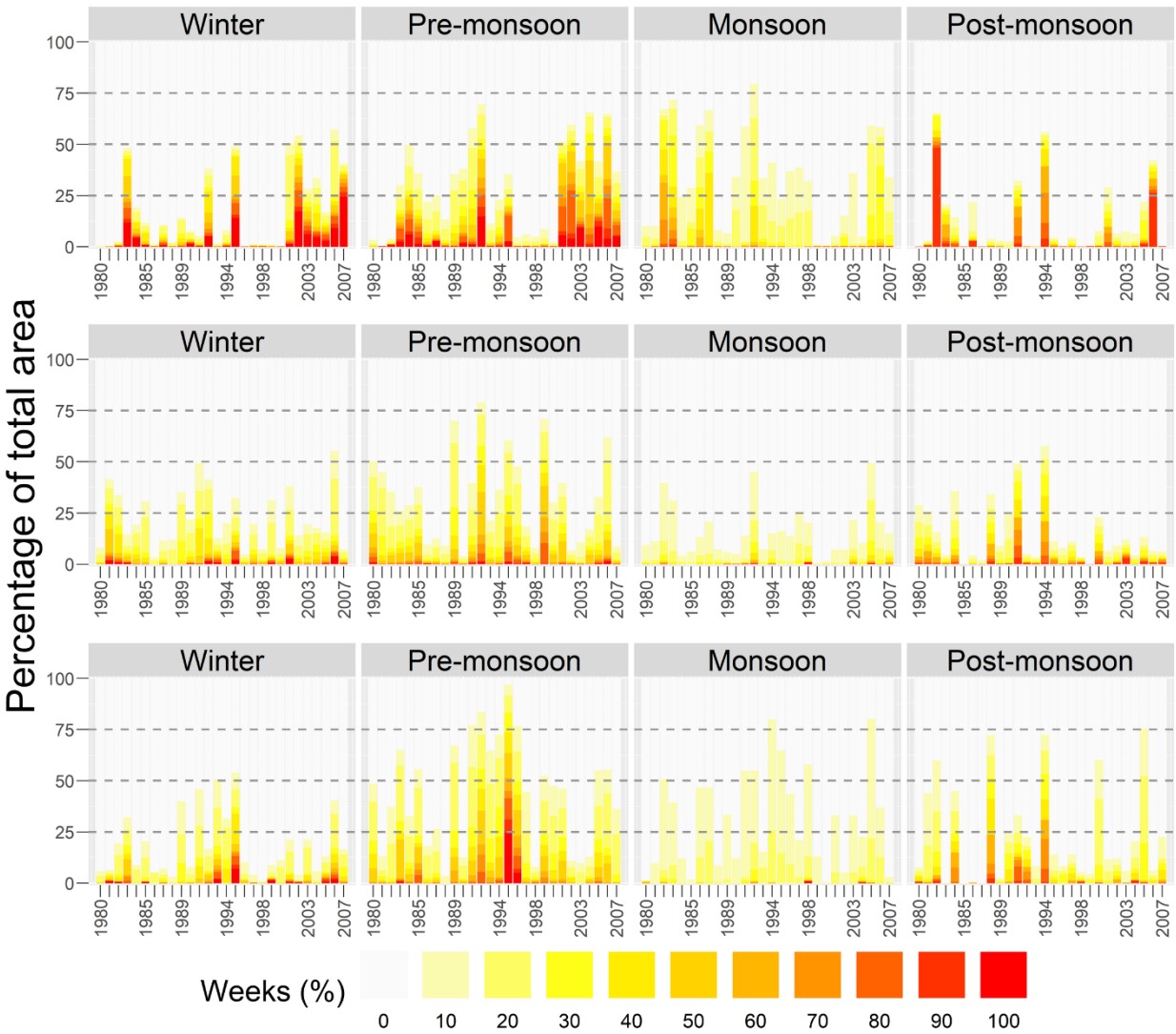


**Figure 12: Percentage of weeks with severe drought in the trans-Himalaya (top), the mountains (middle), and the plains (bottom)**

Note: Each colour band shows the respective HRU's area combined

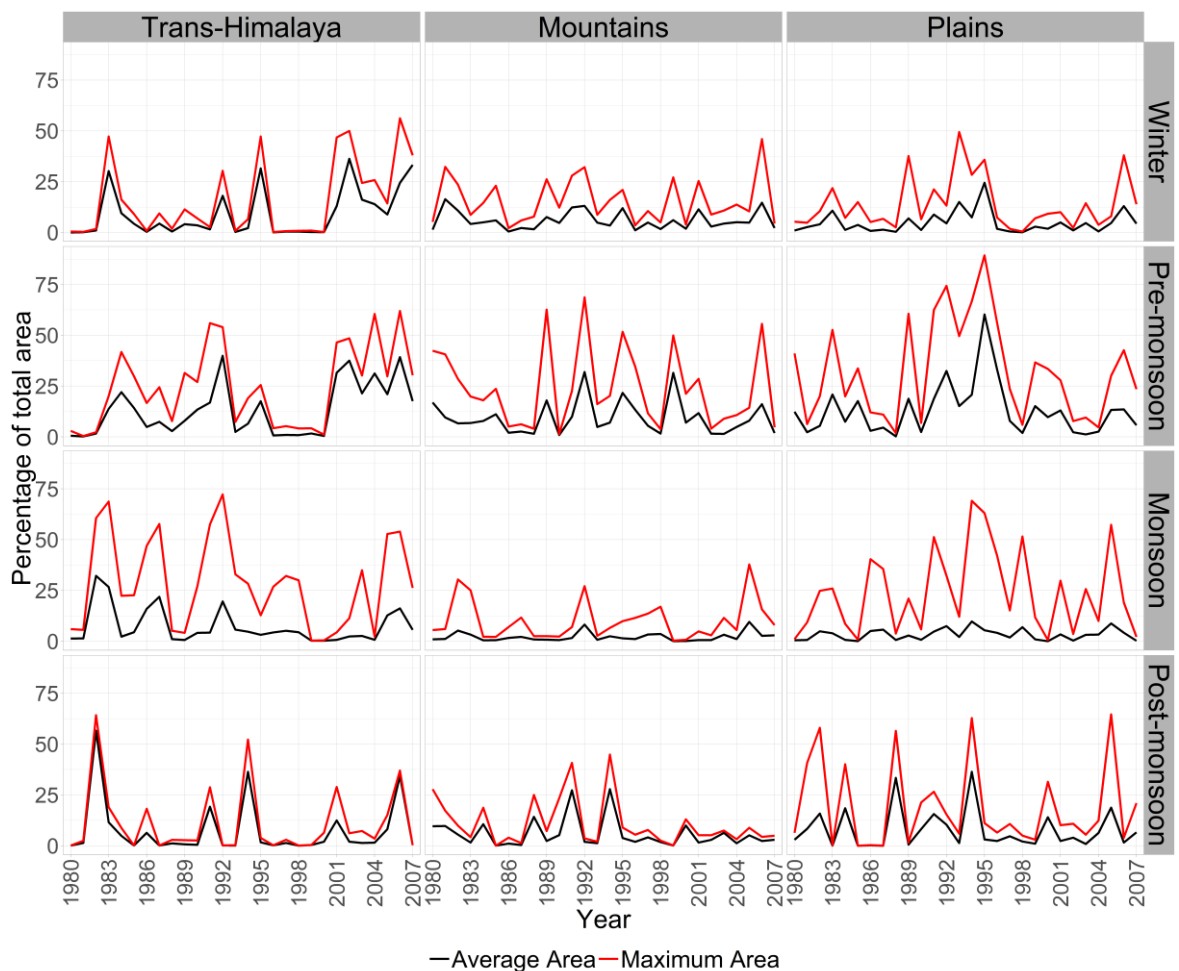

**Figure 13: Area under drought (SMDI values below –3) in the three regions of the KRB**