# Peer review of "Space-time variability of soil moisture droughts in the Himalayan region"

_Hydrology and Earth System Sciences, 2020_

## Referee Comment (RC1) · Anonymous Referee #1 · 3 Nov 2020

The manuscript investigates the spatial and temporal variability of soil moisture droughts in the trans-Himalaya (Tibet), the high and middle mountains (Nepal), and the southern plains of the Koshi river basin (in Nepal and India). In particular, the basin's soil moisture was simulated using the J2000 hydrological model, validated against observed discharge and evapotranspiration. Then the Soil Moisture Drought Index (SMDI) was calculated and compared with the SPI to identify the variation of the drought indication in space and time.

General comments

The paper presents an interesting topic, although it is not totally novel from a methodological point of view. The grammar and language style are satisfactory. The following few points need clarification prior to publication.

[Figure]

Several indices have been developed so far to investigate agricultural droughts (see, for instance, https://nhess.copernicus.org/articles/20/471/2020/nhess-20-471-2020.pdf and references therein). In the introduction, the paper would benefit from a discussion on why SMDI has been preferred to other indices.

SMDI is calculated on a weekly basis in Equation (1). The authors should clarify how the SMDI values can be calculated on a seasonal basis.

According to LL 294-296, SPI values are computed on the same seasonal scale of SMDI (i.e. winter, DJF, pre-monsoon, MAM, monsoon, JJAS, and post-monsoon, ON). However, SPI affecting soil moisture can be related to different aggregation periods. A sensitivity analysis could help to identify the appropriate aggregation period for a better comparison with SMDI.

Technical comments

L18: something is missing after "actual". L36: delete "and" after "hazards". LL 114-115: "SPI is a widely used index ..." moves this sentence before in the introduction. LL 351-352: There is a repetition in these lines. Figures 8 and 9: place the panels vertically rather than horizontally.

---

## Referee Comment (RC2) · Anonymous Referee #2 · 4 Nov 2020

The paper deals with soil moisture drought using an drought index. The soil moisture data were derived from model simulation. The results are reasonable. Some specific comments are as follows.

ïijĹ1ïijĽThe data length is 1980-2007, why the data after 2008 were not used. (2) the model simulated soil moisture was applied to identify soil moisture drought, but does the simulated soil moisture reflect the real soil moisture? Although the hydrological model had a good performance, if there is irrigated area in the study area, does the model consider this condition? (3) The paper analyzed the spatial drought events. I did not see any spatial distribution of the drought events. The authors should show the spatial drought condition using a map. (4) Section 4.2.5, historical incidence of drought. This section should be used to assess the applicability of SMDI. Therefore, it

[Figure]

is better set at the beginning of section 4.2. Moreover, when assessing this index, the onset, duration and termination of the drought should be provided by spatial distribution map. (5) SMDI and SPI were compared and showed some obvious differences. The reasons should be discussed. (6) Discussion is a very important part for a paper, and should be in a separate section.

—————————————————

---

## Referee Comment (RC3) · Anonymous Referee #3 · 4 Nov 2020

The paper aim is an estimation of spatial and temporal variability of a soil moisture deficit index (MDSI) in the transboundary Koshi River basin, located partly in China, India and Nepal. The aim is achieved by applying the process based J2000 hydrological model. The authors follow a standard procedure of calibrating and validating the model using the available observations of precipitation, temperature and flow data at three sub-regions, trans-Himalaya, plains and the mountains. In addition, model estimates of evapotranspiration was validated using the available PET measurements from three meteorological stations located at each region considered.

General comments

The novelty of the paper consists in an application of J2000 model to simulate soil moisture in the basin. The approach is standard and the comparison with observations

is limited to temporal scale, whilst spatial scale is widely discussed. The paper is written reasonably well. The material is interesting to a wide scientific community. The division of the basin into three characteristic sub-regions was very useful and made the discussion of the results more clear. The authors identified 18,557 HRUs within the Koshi River basin. As each HRU requires a number of specific parameters to be identified, there is no surprise, that the model can reproduce well the discharge. Still it is not sure if the identified HRU's give physically sound performance. The authors are asked to give an information on the number of parameters that needed to be set in the model and a number of parameters that were calibrated. A discussion on data quality and uncertainty of the model results should be provided. The discussion also should be extended by a presentation of spatial variability of the resulting soil moisture deficit patterns and their comparison with the SPI and, additionally, SPEI indices for different sub-regions and specific time periods.

Specific comments

Line 301-302: sentence starting with Due to lack of consistency . . . is not necessary here (message repeated further down)

Line 352 should be: Supplementary Figure 1

Line 437: Figure number is missing

Figs 8-10 are not easy to read. It is a pity, as a comparison of those figures gives the answer to the research questions.

[Figure]

---

## Author Comment (AC1) · 27 Nov 2020

**Interactive comments on "Space-time variability of soil moisture droughts in the Himalayan region"**

Santosh Nepal et al.

Santosh.Nepal@icimod.org

We are very grateful to the reviewer #1 for providing valuable comments to our paper. We have greatly benefited from these comments. We hereby provide a detailed response to these comments:

The reviewer comment is marked as **[Comment]** and our response immediately as **[Response]** (in blue font) and part of the revision in the manuscript in *italics*.

**General comments**

**[Comment]** Several indices have been developed so far to investigate agricultural droughts (see, for instance, https://nhess.copernicus.org/articles/20/471/2020/nhess-20-471-2020.pdf and references therein). In the introduction, the paper would benefit from a discussion on why SMDI has been preferred to other indices.

**[Response]**

We have elaborated the benefit of SMDI over other indices. We also referred to *Monteleone et al. 2020 in the introduction.*

The revised paragraph in the introduction reads as (the yellow highlights are added text):

*There are many drought indices, such as the Standardized Precipitation Index (SPI), Standardized Precipitation Evaporation Index (SPEI), Evapotranspiration Deficit Index (ETDI), Soil Moisture Deficit Index (SMDI), Aggregate Drought Index (ADI), Standardized Runoff Index (SRI), Probabilistic Precipitation Vegetation Index (PPVI) and Palmer Drought Severity Index (PDSI), which indicates the differential nature of droughts that might occur at different time intervals and lag times (Bayissa et al., 2018; Huang et al., 2015; Narasimhan and Srinivasan 2005; Monteleone et al. 2020). Focusing on soil moisture variability, SMDI takes into account more variables (such as evapotranspiration, soil properties, and root depth) than SPI and SPEI, which takes into account precipitation, and precipitation & evapotranspiration, respectively. Therefore, SMDI can provide dependable information to interpret the occurrence and severity of the agricultural drought. Similarly, SPI is a widely used index to characterise meteorological droughts on a range of timescales. Monteleone et al. (2020) suggested list of indices for agriculture drought monitoring, including Evapotranspiration Deficit Index (ETDI), Normalized Difference Vegetation Index (NDVI), Soil Moisture Anomaly Index (SMAI), SPI, SMDI and Standardized Soil Moisture Index (SSI). Each of these indices has their own pros and cons for different climatic variables they use for drought calculation, data requirement and availability and their potential use for agricultural drought monitoring.*

*This paper aims at assessing soil moisture droughts in the Koshi River Basin. To understand soil moisture droughts, this paper considered the Soil Moisture Deficit Index (SMDI)* *and Standardised Precipitation Index (SPI) for 28 years (1980–2007).* *For soil moisture variability, SMDI takes into account precipitation, temperature, evaporation, soil and vegetation properties affecting soil moisture conditions.* *For this purpose, the basin's soil moisture was simulated with the use of the process-based J2000 hydrological model, which was validated against observed discharge and evapotranspiration. The J2000 model has been successfully used to investigate hydrological droughts in Central Vietnam (Firoz et al., 2018; Nauditt et al., 2017). This paper specifically investigates the spatial and temporal variability of soil moisture for the trans-Himalaya (Tibet), the high and middle mountains (Nepal), and the southern plains of the river basin (in Nepal and India). We also compared the SMDI with the SPI to identify the variation of the drought indication in space and time. SPI is a widely used index to characterise meteorological droughts on a range of timescales. To the best of our knowledge, soil moisture drought is being studied for the first time in the transboundary Koshi River basin and this paper provides insights into its spatio-temporal variability in the historic time period under consideration.*

**[Comment]**

SMDI is calculated on a weekly basis in Equation (1). The authors should clarify how the SMDI values can be calculated on a seasonal basis.

**[Response]**

We have revised the paragraph in the method section to elaborate on this aspect.

The revised paragraph in the method section reads as:

*The calculation of the SMDI has been implemented in the JAMS modelling system using two individual JAMS components, namely SMDI_DataCollect and SMDI_Calc. The first component is used to collect soil moisture data for each HRU during the normal hydrological simulation with J2000. In addition, this component also calculates long-term soil water statistics for each HRU (for example, $MSW_w$). Once this is finished, the second component (SMDI_Calc) will calculate the SMDI values for each HRU based on their weekly soil moisture values ($SW_{y,w}$) and long-term statistics ($MSW_w$, $minSW_w$, $maxSW_w$). While weekly intervals are used as the default, the component can calculate SMDI values based on any given aggregation period, for example, to consider individual characteristics of specific vegetation types. As described above, the HRUs were segregated into three geographical regions, trans-Himalaya, mountains, and plains, as the climatic conditions are different in each of these zones. Similarly, the SMDI values were analysed separately for four seasons: monsoon (June–September), post-monsoon (October–November), winter (December–February), and pre-monsoon (March–May). Since these seasons are defined based on variations in precipitation and temperature, the SMDI is calculated for these seasons to track the variation caused by these meteorological drivers.* *For this, we averaged the weekly SMDI values for a given season. In this way, the dominating climatic characteristics are maintained at the seasonal level.*

**[Comment]**

According to LL 294-296, SPI values are computed on the same seasonal scale of SMDI (i.e. winter, DJF, pre-monsoon, MAM, monsoon, JJAS, and post-monsoon, ON). However, SPI affecting soil moisture can be related to different aggregation periods. A sensitivity analysis could help to identify the appropriate aggregation period for a better

comparison with SMDI.

**[Response]**

We agree with the reviewer that SPI affecting soil moisture can be related to different aggregation periods. In our paper, our aim was not to correlate the aggregation period for SPI and SMDI. The focus of the paper was to look at soil moisture variability and also if SPI was able to explain that variability. The aim of using SPI was rather to show that variation in drought indication by SPI (which considers precipitation only) and SMDI (which considers more variables than precipitation, such as temperature, evaporation, soil and vegetation conditions) at the seasonal scales. Here, the seasonal aggregation is more logical because of the four dominating and distinct seasons in the KRB based on precipitation and temperature condition. Therefore, a sensitivity analysis to identify the appropriate aggregation period is out of the scope of this study.

We also clarified the SPI and SMDI aggregation period in the method section 3.3.3, which reads as:

*The Standardized Precipitation Index (SPI) is the most commonly used indicator for detecting and characterising meteorological drought on different timescales. We calculated the seasonal SPI which was implemented as a JAMS component. The SPI is calculated based on a long-time series of precipitation data. The SPI measures precipitation anomalies based on a comparison of observed total precipitation amounts for an accumulation period (for example, 1, 3, 12, or 48 months) with the long-term historic record for that period. The probability distribution of the historic record was fitted to a gamma distribution, which was then transferred to a normal distribution to get a mean SPI value of zero (McKee et al., 1993; McKee et al., 1995). To compare the seasonal SMDI with the SPI, we calculated the SPI data for the same period of four seasons used to calculate the SMDI. For this, the aggregation period was based on the end month of each season and SPI accumulation period was chosen based on the months. For winter, 3 months SPI was calculated for the month of February; for pre-monsoon, 3 months SPI for May; for monsoon, 4 months SPI for September and for post-monsoon, 2 months SPI for November. In this manner, the occurrence of drought based on the SPI and SMDI in different time intervals was discussed together.*

**[Technical comments]**

**L18:** something is missing after "actual".
Corrected:
The revised sentence reads as:

*In order to identify drought conditions based on the simulated soil moisture, the Soil Moisture Deficit Index (SMDI) was then calculated, considering the derivation of actual soil moisture long-term soil moisture on a weekly timescale.*

**L36:** delete "and" after "hazards".
Corrected:

The revised sentence reads as:

*Droughts are considered one of the world's major social and economic hazards, which have been increasing in recent decades.*

**LL 114-115:** "SPI is a widely used index : : :" moves this sentence before in the introduction.

We moved the sentence before the introduction:

LL 351-352: There is a repetition in these lines.

The repeated lines are deleted.

**Figures 8 and 9:** place the panels vertically rather than horizontally.

The panels are replaced. Please see below the revised maps arranged vertically with new figure numbers

[Figure]

**Figure 9 Spatial and seasonal variability of the SMDI in trans-Himalaya (top), the mountains (middle), and the plains (bottom)**
Note: Each colour band shows the respective HRU's area combined.

[Figure]

**Figure 10 Spatial and seasonal variability of the SPI in the trans-Himalaya (top), the mountains (middle), and the plains (bottom)**

Note: Each colour band shows the respective HRU's area combined.

[Figure]

**Figure 12: Percentage of weeks with severe drought in the trans-Himalaya (top), mountains (middle), and plains (bottom)**
Note: Each colour band shows the respective HRU's area combined

---

## Author Comment (AC2) · 27 Nov 2020

**Interactive comments on "Space-time variability of soil moisture droughts in the Himalayan region"**

Santosh Nepal et al.

Santosh.Nepal@icimod.org

**Anonymous Referee #2**

We are very grateful to the reviewer #2 for providing valuable comments to our paper. We have greatly benefited from these comments. We hereby provide a detailed response to these comments:

The reviewer comment is marked as **[Comment]** and our response immediately as **[Response]** (in blue font) and part of the revision in the manuscript in *italics*.

**[Comment]**
The data length is 1980-2007, why the data after 2008 were not used.

**[Response]**
We used the APHRODITE datasets in the data-scarce region of the northern part of the Koshi basin in combination with other datasets. APHRODITE data (version: V1101) is only available up to 2007. Because of this reason, we limited the analysis period up to 2007 (i.e. 1980-2007: 28 years analysis period, 1979 as warm up year).

We have revised the paragraph to clarify the data limitation in uncertainty section in Discussion.

*Uncertainties and limitations*

*The model results are subject to several uncertainties and limitations which are briefly described below. The calibration and validation of hydrological model results are subject to uncertainty arising from model input data, parameter and structural uncertainty. In the mountainous region, the representation of the observed station network is sparse and limited which is the case in the KRB. For the northern part of China and southern Indian side, gridded datasets were used compared to station data in the Southern Himalaya in Nepal part. The application of APHRODITE data in the northern region has limited the study period up to 2007 because of the data available only up to this period. The station data are mostly limited to the lower elevation areas with limited station network in high altitude areas. Both the gridded and observation network have their advantage and disadvantages for modelling applications. Nonetheless, our approach of using both gridded and station data along with discharge data have enabled us to use the modelling period of 28 years which is a relatively a longer period in the case of transboundary KRB.*

*Regarding the parameter uncertainty, the application of the J2000 hydrological model in the previous studies has shown the potential of spatial transferability of model parameters within the sub-catchments of the Koshi basin (Nepal et al. 2017). The generalized likelihood uncertainty estimation (GLUE) analysis of two sub-catchments of the Koshi basin suggests that most of the time the parameter uncertainty can be explained within the ensemble range of multiple simulations, except some flood events. Supplementary Table 3 shows the J2000 model parameters including the selected parameters which were used for sensitivity and uncertainty analysis by Nepal et al. (2017). Similarly, there were good matches on the category of high, moderate and low sensitive parameters between the two catchments suggesting the robustness of the model in the Himalayan catchments. The results have also suggested that spatial transferability of model parameters in the neighbouring catchment with similar climatic and hydrological conditions are possible in the Himalayan region (Nepal et al, 2017), however, some variation is parameters can be expected if the scale of the basin size and climatic conditions differ (Eeckman et al. 2019; Shrestha and Nepal, 2019). Besides, the soil moisture from the J2000 model was not validated independently due to the lack of the observed data and validation was only limited to discharge and evaporation*

*data. Despite these uncertainties and limitation, the model has replicated overall hydrological behaviour including both low and high flows, similar to the previous studies in the Koshi basin (Nepal et al. 2014, Nepal et al, 2017; Eeckman et al. 2019).*

**[Comment]**
The model simulated soil moisture was applied to identify soil moisture drought, but does the simulated soil moisture reflect the real soil moisture? Although the hydrological
model had a good performance, if there is irrigated area in the study area, does the
model consider this condition?

**[Response]**
Our model has not considered irrigation systems. We have validated our results with discharge and evaporation data. Due to the lack of soil moisture data in the study area, we could not validate the simulated soil moisture directly . Therefore we have assumed that as the multi-response outcomes (Q and ET) from the model have been validated, simulated soil moisture should be a good representation of the basin response. We have made this limitation clear in the method section in the revised version now.

The revised sentence in the method section (last paragraph) reads as:

*Overall, the soil moisture conditions can be influenced by irrigation in plain areas of Terai. We have not considered irrigation and artificial water storage while setting up the model. In those areas, the supplemental irrigation might have elevated the soil moisture level in irrigated fields. Similarly, the soil moisture derived from the model was not validated independently due to the lack of the observed data and validation was only limited to discharge and evaporation data.*

**[Comment]**
The paper analyzed the spatial drought events. I did not see any spatial distribution of the drought events. The authors should show the spatial drought condition using a map.

**[Response]**

Thank you for pointing out the important aspects of spatial maps. Now we have included spatial drought maps in two places. Figure 8 shows the average spatial SMDI maps including the driest and wettest year. Similarly, Figure 11 shows the duration of drought events (i.e. SMDI below -3.0), including the driest and wettest year. To complement the spatial maps, we also calculated average SMDI values and duration of drought events for 3 physiographic region and whole basin for each year (Table 3 and Table 4), and also discussed these aspects in results and discussion section.

[Figure]

Figure 8: Spatial maps of average annual SMDI (1980-2007), dryest year (1992) and wettest year (1998).

[Figure]

Figure 11: Spatial maps of average duration of drought i.e. SMDI < -3.0, dryest year (1992) and wettest year (1998).

Table 3: Average annual SMDI values from 1980-2007 for Trans-Himalaya, Mountains and Plains and for the whole Koshi basin. The red bar shows the negative and blue shows the positive SMDI values, the average SMDI values for each year are given in the respective rows.

[Figure]

| Year | Trans-Himalaya | Mountains | Plains | Koshi Basin |
|---|---|---|---|---|
| 1980 | 1.49 | 0.21 | 0.57 | 0.72 |
| 1981 | 1.26 | -0.20 | 0.28 | 0.4 |
| 1982 | -0.66 | -0.23 | -0.32 | -0.39 |
| 1983 | -1.21 | 0.45 | -0.04 | -0.21 |
| 1984 | -0.45 | 0.07 | 0.31 | -0.02 |
| 1985 | -0.39 | 0.15 | 0.51 | 0.09 |
| 1986 | -0.29 | 0.96 | 0.18 | 0.33 |
| 1987 | -0.33 | 0.55 | 0.65 | 0.31 |
| 1988 | 0.35 | 0.09 | 0.11 | 0.18 |
| 1989 | -0.29 | 0.04 | -0.17 | -0.13 |
| 1990 | -0.50 | 0.35 | 0.01 | -0.02 |
| 1991 | -0.49 | -0.57 | -0.42 | -0.5 |
| 1992 | -1.08 | -0.78 | -0.91 | -0.91 |
| 1993 | -0.24 | 0.34 | -0.24 | -0.02 |
| 1994 | -1.11 | -0.31 | -0.97 | -0.76 |
| 1995 | -1.00 | -0.30 | -0.98 | -0.72 |
| 1996 | 1.06 | 0.41 | -0.01 | 0.49 |
| 1997 | 0.12 | 0.00 | -0.17 | -0.02 |
| 1998 | 0.92 | 0.71 | 0.87 | 0.82 |
| 1999 | 1.18 | -0.11 | 0.36 | 0.43 |
| 2000 | 1.29 | 0.00 | 0.22 | 0.47 |
| 2001 | -0.84 | -0.09 | -0.28 | -0.38 |
| 2002 | -1.23 | 0.35 | 0.41 | -0.12 |
| 2003 | -0.79 | 0.25 | 1.01 | 0.16 |
| 2004 | -0.87 | 0.43 | 0.24 | -0.03 |
| 2005 | -1.05 | -0.23 | -0.72 | -0.63 |
| 2006 | -1.35 | -0.26 | -0.41 | -0.64 |
| 2007 | -0.28 | 0.62 | 0.39 | 0.27 |

Table 4: Duration of drought events for trans-Himalaya, Mountains and Plains and for the whole Koshi basin. The red bars corresponding to values on the rows show the number of weeks where SMDI < -3.0.

| Year | Trans-Himalaya | Mountains | Plains | Koshi Basin |
|---|---|---|---|---|
| 1980 | 0.3 | 3.3 | 2.1 | 2.0 |
| 1981 | 0.4 | 4.3 | 1.4 | 2.2 |
| 1982 | 10.6 | 3.7 | 3.3 | 5.7 |
| 1983 | 11.5 | 2.2 | 5.1 | 6.0 |
| 1984 | 5.0 | 2.6 | 2.8 | 3.4 |
| 1985 | 3.2 | 2.4 | 2.9 | 2.8 |
| 1986 | 4.1 | 0.7 | 1.4 | 2.0 |
| 1987 | 5.7 | 1.1 | 1.8 | 2.7 |
| 1988 | 0.7 | 1.7 | 2.9 | 1.8 |
| 1989 | 1.7 | 3.7 | 4.0 | 3.2 |
| 1990 | 3.1 | 1.3 | 1.3 | 1.8 |
| 1991 | 4.7 | 5.4 | 5.8 | 5.3 |
| 1992 | 11.1 | 7.6 | 7.1 | 8.5 |
| 1993 | 1.4 | 1.5 | 4.6 | 2.4 |
| 1994 | 5.0 | 4.0 | 8.5 | 5.7 |
| 1995 | 7.2 | 5.0 | 12.8 | 8.0 |
| 1996 | 0.9 | 2.2 | 5.7 | 2.9 |
| 1997 | 1.2 | 2.4 | 1.7 | 1.8 |
| 1998 | 1.0 | 1.3 | 1.4 | 1.2 |
| 1999 | 0.3 | 4.9 | 2.5 | 2.8 |
| 2000 | 0.3 | 2.0 | 2.7 | 1.7 |
| 2001 | 6.9 | 3.3 | 3.3 | 4.4 |
| 2002 | 10.4 | 0.9 | 0.9 | 3.8 |
| 2003 | 5.5 | 1.9 | 1.4 | 2.9 |
| 2004 | 6.2 | 1.6 | 1.6 | 3.0 |
| 2005 | 6.8 | 3.8 | 5.6 | 5.3 |
| 2006 | 13.9 | 4.7 | 4.2 | 7.4 |
| 2007 | 7.8 | 1.3 | 1.9 | 3.5 |

**[Comment]**
Section 4.2.5, historical incidence of drought. This section should be used to assess the applicability of SMDI. Therefore, it is better set at the beginning of section 4.2.

**[Response]**
Thanks for pointing out this important aspect. In combination with reviewer #2 (last comment) and #3 comments, we have now developed a new section 'Discussion' after 'Result'. Along with new information on discussion, we also highlighted three aspects in discussion: 1) Discussion related to results 2) uncertainty and limitation  and 3) historical incidences of drought

The new discussion section reads as:

[revised manuscript text omitted]

**[Comment]**
Moreover, when assessing this index, the onset, duration and termination of the drought should be provided by spatial distribution map.

**[Response]**

We have discussed the magnitude of 'drought' events (i.e. SMDI values lower than -3.0). Now we have included spatial drought maps in two places. Figure 8 shows the average spatial SMDI maps including the driest and wettest year. Similarly, Figure 11 shows the duration of drought events (i.e. SMDI below -3.0), including the driest and wettest year. To complement this figure, we also calculated average SMDI values and duration of drought events for 3 physiographic region and whole basin for each year (Table 3 and Table 4), and also discussed these aspects in results and discussion section. (The figures and tables are already provided in connection to the earlier comment).

However, the study of onset and termination of spatial drought will need the development of its own robust methodology. We think this warrants a study of its own and is out of the scope of our study. More specifically, to show the onset and termination of SMDI, we would need to look at the soil moisture values at weekly scale while our aggregation period for this paper is the seasonal scale. Therefore, the onset and termination would be unsuitable to incorporate within the existing methodological approach of this paper, and not included in the revised version.

**[Comment**

(5) SMDI and SPI were compared and showed some obvious differences. The reasons should be discussed.

**[Response]**

To address this comment, we have added a paragraph in the discussion section and discussed the differences in SMDI and SPI. The paragraph in the discussion section reads as:

*Analysis related to SMDI and SPI, the former is able to reflect variations in soil moisture conditions better than SPI which shows normal conditions. As shown in trans-Himalaya, the period after 2001 when SPI shows wetness and SMDI show dryness during the pre-monsoon. It is because SMDI incorporates additional variables (temperature, evaporation, vegetation, root depth, and soil water holding capacities) to calculate soil moisture variability compared to only precipitation variables by SPI. As expected, the SPI gives a more homogeneous response because of the lack of the representation of physiographic differences. An example of this behaviour can be seen in winter 2006 where SPI indicates a severe drought in over 80 % of the area of trans-Himalaya and mountains (Figure 9). In contrast, SMDI shows a more differentiated pattern (Figure 8) where during winter drought conditions are indicated for roughly half of the area with severe values only for 10 to 20% of the area. Most likely, one reason for this more differentiated picture is the consideration of soil water storage in the SMDI. The remaining soil water after the post-monsoon can be very important for the water supply and overshadow the effect of (missing) precipitation in winter. Additionally, this effect can be amplified by the low ET volumes during winter (Figure 3) that deplete the stored soil moisture only slowly, resulting in higher SMDI values. The shown differences in the SMDI are caused by varying soil water storage capacities which control the duration of periods during which higher SMDI can be maintained without precipitation. The years for which both SPI and SMDI show matching drought conditions can be mainly attributed to them being the lowest rainfall periods (Figure 5).*

**[Comment]**

(6) Discussion is a very important part for a paper, and should be in a separate section.

**[Response]**

In relation to an earlier comment, now we have developed a separate section for 'Discussion' and highlighted three aspects: 1) Discussion related to results 2) uncertainty and limitation and 3) historical incidences of drought

The new discussion section is already pasted above in relation to the earlier comment:

---

## Author Comment (AC3) · 27 Nov 2020

**Interactive comments on "Space-time variability of soil moisture droughts in the Himalayan region"**

Santosh Nepal et al.

Santosh.Nepal@icimod.org

**Anonymous Referee #3**

We are very grateful to the reviewer #3 for providing valuable comments to our paper. We have greatly benefited from these comments. We hereby provide a detailed response to these comments:

The reviewer comment is marked as **[Comment]** and our response immediately as **[Response]** (in blue font) and part of the revision in the manuscript in *italics*.

**[Comment]**
The approach is standard and the comparison with observations is limited to temporal scale, whilst spatial scale is widely discussed.

**[Response]**
Thank you for bringing this important point about the spatial maps. Now we have included spatial maps to show the average SMDI and duration of drought events:

We have discussed the magnitude of 'drought' events (i.e. SMDI values lower than -3.0). Now we have included spatial drought maps in two places. Figure 8 shows the average spatial SMDI maps including the driest and wettest year. Similarly, Figure 11 shows the duration of drought events (i.e. SMDI below -3.0), including the driest and wettest year. To complement this figure, we also calculated average SMDI values and duration of drought events for 3 physiographic region and whole basin for each year (Table 3 and Table 4), and also discussed these aspects in results and discussion section.

The new addition of spatial maps and tables are provided below:

[Figure]

Figure 8: Spatial maps of average annual SMDI (1980-2007), dryest year (1992) and wettest year (1998).

[Figure]

Figure 11: Spatial maps of average duration of drought i.e. SMDI < -3.0, dryest year (1992) and wettest year (1998).

Table 3: Average annual SMDI values from 1980-2007 for Trans-Himalaya, Mountains and Plains and for the whole Koshi basin. The red bar shows the negative and blue shows the positive SMDI values, the average SMDI values for each year are given in the respective rows.

[Figure]

| Year | Trans-Himalaya | Mountains | Plains | Koshi Basin |
|---|---|---|---|---|
| 1980 | 1.49 | 0.21 | 0.57 | 0.72 |
| 1981 | 1.26 | -0.20 | 0.28 | 0.4 |
| 1982 | -0.66 | -0.23 | -0.32 | -0.39 |
| 1983 | -1.21 | 0.45 | -0.04 | -0.21 |
| 1984 | -0.45 | 0.07 | 0.31 | -0.02 |
| 1985 | -0.39 | 0.15 | 0.51 | 0.09 |
| 1986 | -0.29 | 0.96 | 0.18 | 0.33 |
| 1987 | -0.33 | 0.55 | 0.65 | 0.31 |
| 1988 | 0.35 | 0.09 | 0.11 | 0.18 |
| 1989 | -0.29 | 0.04 | -0.17 | -0.13 |
| 1990 | -0.50 | 0.35 | 0.01 | -0.02 |
| 1991 | -0.49 | -0.57 | -0.42 | -0.5 |
| 1992 | -1.08 | -0.78 | -0.91 | -0.91 |
| 1993 | -0.24 | 0.34 | -0.24 | -0.02 |
| 1994 | -1.11 | -0.31 | -0.97 | -0.76 |
| 1995 | -1.00 | -0.30 | -0.98 | -0.72 |
| 1996 | 1.06 | 0.41 | -0.01 | 0.49 |
| 1997 | 0.12 | 0.00 | -0.17 | -0.02 |
| 1998 | 0.92 | 0.71 | 0.87 | 0.82 |
| 1999 | 1.18 | -0.11 | 0.36 | 0.43 |
| 2000 | 1.29 | 0.00 | 0.22 | 0.47 |
| 2001 | -0.84 | -0.09 | -0.28 | -0.38 |
| 2002 | -1.23 | 0.35 | 0.41 | -0.12 |
| 2003 | -0.79 | 0.25 | 1.01 | 0.16 |
| 2004 | -0.87 | 0.43 | 0.24 | -0.03 |
| 2005 | -1.05 | -0.23 | -0.72 | -0.63 |
| 2006 | -1.35 | -0.26 | -0.41 | -0.64 |
| 2007 | -0.28 | 0.62 | 0.39 | 0.27 |

Table 4: Duration of drought events for trans-Himalaya, Mountains and Plains and for the whole Koshi basin. The red bars corresponding to values on the rows show the number of weeks where SMDI < -3.0.

| Year | Trans-Himalaya | Mountains | Plains | Koshi Basin |
|---|---|---|---|---|
| 1980 | 0.3 | 3.3 | 2.1 | 2.0 |
| 1981 | 0.4 | 4.3 | 1.4 | 2.2 |
| 1982 | 10.6 | 3.7 | 3.3 | 5.7 |
| 1983 | 11.5 | 2.2 | 5.1 | 6.0 |
| 1984 | 5.0 | 2.6 | 2.8 | 3.4 |
| 1985 | 3.2 | 2.4 | 2.9 | 2.8 |
| 1986 | 4.1 | 0.7 | 1.4 | 2.0 |
| 1987 | 5.7 | 1.1 | 1.8 | 2.7 |
| 1988 | 0.7 | 1.7 | 2.9 | 1.8 |
| 1989 | 1.7 | 3.7 | 4.0 | 3.2 |
| 1990 | 3.1 | 1.3 | 1.3 | 1.8 |
| 1991 | 4.7 | 5.4 | 5.8 | 5.3 |
| 1992 | 11.1 | 7.6 | 7.1 | 8.5 |
| 1993 | 1.4 | 1.5 | 4.6 | 2.4 |
| 1994 | 5.0 | 4.0 | 8.5 | 5.7 |
| 1995 | 7.2 | 5.0 | 12.8 | 8.0 |
| 1996 | 0.9 | 2.2 | 5.7 | 2.9 |
| 1997 | 1.2 | 2.4 | 1.7 | 1.8 |
| 1998 | 1.0 | 1.3 | 1.4 | 1.2 |
| 1999 | 0.3 | 4.9 | 2.5 | 2.8 |
| 2000 | 0.3 | 2.0 | 2.7 | 1.7 |
| 2001 | 6.9 | 3.3 | 3.3 | 4.4 |
| 2002 | 10.4 | 0.9 | 0.9 | 3.8 |
| 2003 | 5.5 | 1.9 | 1.4 | 2.9 |
| 2004 | 6.2 | 1.6 | 1.6 | 3.0 |
| 2005 | 6.8 | 3.8 | 5.6 | 5.3 |
| 2006 | 13.9 | 4.7 | 4.2 | 7.4 |
| 2007 | 7.8 | 1.3 | 1.9 | 3.5 |

**[Comment]**
The authors are asked to give an information on the number of parameters that needed to be set in the model and a number of parameters that were calibrated.

**[Response]**
The table of the model parameters and their range are provided in supplementary table 3

Supplementary Table 1: Calibration parameters in the J2000 hydrological model.

Note: the parameters (in bold) were the 16 selected parameters for sensitivity and uncertainty analysis by Nepal et al. (2017).

| Parameter | Description | Calibrated value | Normal range | Units |
|---|---|---|---|---|
| *Precipitation distribution* | | | | |
| Trs | Base temperature | 0 | −1 to +1 | °C |
| Trans | Parameter range for mixed rain and snow | 2 | −2 to +2 | °C |
| *Interception module* | | | | |
| a_rain | Interception storage for rain | 1 | 0–5 | mm |
| a_snow | Interception storage for snow | 1.28 | 0–5 | mm |
| *Snow module* | | | | |
| CritDens | Critical density of snow | 0.381 | 0–1 | % |
| ColdContent | Cold content of snowpack | 0.0012 | 0–1 | NA |
| **BaseTemp** | Threshold temperature for snowmelt | 0 | −5 to +5 | °C |
| **Tfactor** | Melt factor by sensible heat | 2.84 | 0–5 | NA |
| Rfactor | Melt factor by liquid precipitation | 0.21 | 0–5 | NA |
| Gfactor | Melt factor by soil heat flow | 3.73 | 0–5 | NA |
| *Glacier module* | | | | |
| **meltFactorIce** | Melt factor for ice melt | 0.5 | 0–5 | NA |
| alphaIce | Radiation melt factor for ice | 0.1 | 0–5 | NA |
| kIce | Routing coefficient for ice melt | 15 | 0–50 | NA |
| kSnow | Routing coefficient for snowmelt | 10 | 0–50 | NA |
| kRain | Routing coefficient for rainfall–run-off | 5 | 0–50 | NA |
| debrisFactor | Debris factor for ice melt | 5 | 0–10 | NA |
| **glacierTbase** | Threshold temperature for snowmelt | −1 | −5 to +5 | °C |

| | | | | |
|---|---|---|---|---|
| *Soil module* | | | | |
| soilMaxDPS | Maximum depression storage | 2 | 0–10 | mm |
| **soilLinRed** | Linear reduction coefficient for actual evaporation | 0.6 | 0–1 | |
| **soilMaxInfSummer** | Maximum infiltration in summer | 45 | 0–200 | mm |
| **soilMaxInfWinter** | Maximum infiltration in winter | 50 | 0–200 | mm |
| soilMaxInfSnow | Maximum infiltration in snow-covered areas | 40 | 0–200 | mm |
| soilInpLT80 | Infiltration for areas less than 80% sealing | 0.5 | 0–1 | NA |
| SoilDistMPSLPS | MPS–LPS distribution coefficient | 0.27 | 0–10 | NA |
| SoilDiffMPSLPS | MPS–LPS diffusion coefficient | 0.1 | 0–10 | NA |
| soilOutLPS | Outflow coefficient for LPS | 7 | 0–10 | NA |
| **soilLatVertLPS** | Lateral vertical distribution coefficient | 0.05 | 0–10 | NA |
| **soilMaxPerc** | Maximum percolation rate to groundwater | 30 | 0–100 | mm |
| **soilConcRD1Flood** | Recession coefficient for flood event | 1.1 | 1–10 | NA |
| soilConcRD1Flood threshold | Threshold value for soilConcRD1Flood | 500 | 0–500 | NA |
| **soilConcRD1** | Recession coefficient for overland flow | 1.5 | 1–10 | NA |
| **SoilConcRD2** | Recession coefficient for interflow | 1.8 | 1–10 | NA |
| *Groundwater module* | | | | |
| **gwRG1RG2dist** | RG1–RG2 distribution coefficient | 20 | 0–5 | NA |
| **gwRG1Fact** | Adaptation factor for RG1 flow | 0.05 | 0–10 | NA |
| **gwRG2Fact** | Adaptation factor for RG2 flow | 0.18 | 0–10 | NA |
| gwCapRise | Capillary rise coefficient | 0.01 | 0–10 | NA |
| *Reach routing* | | | | |
| **flowRouteTA** | Flood routing coefficient | 30 | 0–100 | NA |

**[Comment]**

A discussion on data quality and uncertainty of the model results should be provided.

**[Response]**
A new paragraph is added describing the uncertainty and limitations of data quality and availability in the Discussion section. The data description is provided in Supplementary Table 1.

The new paragraph in the discussion section reads as:

*Uncertainties and limitations*

[revised manuscript text omitted]

**[Comment]**

The discussion also should be extended by a presentation of spatial variability of the resulting soil moisture deficit patterns and their comparison with the SPI and, additionally, SPEI indices for different sub-regions and specific time periods.

**[Response]**

Thank you for the important comment on the spatial maps of SMDI. We have now added the spatial maps showing the variation in SMDI and duration of drought events. The maps are posted above in relation to the first comment. The discussion of spatial maps is also included in the results and discussion section.

The main focus of the paper is to understand the soil moisture variability using SMDI index. Additionally, we calculated SPI as a meteorological drought index to see how it differs from soil moisture variability. Besides, we believe that since SPEI also uses precipitation, evaporation and temperature data, the response of SPEI and SMDI might be similar. Comparison of different drought indices would require a huge effort.

The SMDI and SPI comparison are discussed in detail in the 'Discussion' section. The related paragraph reads as:

*Analysis related to SMDI and SPI, the former is able to reflect variations in soil moisture conditions better than SPI which shows normal conditions. As shown in trans-Himalaya, the period after 2001 when SPI shows wetness and SMDI show dryness during the pre-monsoon. It is because SMDI incorporates additional variables (temperature, evaporation, vegetation, root depth, and soil water holding capacities) to calculate soil moisture variability compared to only precipitation variables by SPI. As expected, the SPI gives a more homogeneous response because of the lack of the representation of physiographic differences. An example of this behaviour can be seen in winter 2006 where SPI indicates a severe drought in over 80 % of the area of trans-Himalaya and mountains (Figure 9). In contrast, SMDI shows a more differentiated pattern (Figure 8) where during winter drought conditions are indicated for roughly half of the area with severe values only for 10 to 20% of the area. Most likely, one reason for this more differentiated picture is the consideration of soil water storage in the SMDI. The remaining soil water after the post-monsoon can be very important for the water supply and overshadow the effect of (missing) precipitation in winter. Additionally, this effect can be amplified by the low ET volumes during winter (Figure 3) that deplete the stored soil moisture only slowly, resulting in higher SMDI values. The shown differences in the SMDI are caused by varying soil water storage capacities which control the duration of periods during which higher SMDI can be maintained without precipitation. The years for which both SPI and SMDI show matching drought conditions can be mainly attributed to them being the lowest rainfall periods (Figure 5).*

**Specific comments:**

Line 301-302: sentence starting with Due to lack of consistency : : : is not necessary here (message repeated further down)

Removed

Line 352 should be: Supplementary Figure 1

The supplementary figure numbers are corrected now:

Supplementary Figure 1: Conceptual layout of the J2000 hydrological model

Supplementary Figure 2: Variation in weekly soil moisture for the Koshi River basin, 1980–2007

Line 437: Figure number is missing

Corrected

Figs 8-10 are not easy to read. It is a pity, as a comparison of those figures gives the answer to the research questions

Response:

We have now revised the figure suitable for the journal format. I think it can be read easily now. Please find the figures below:

[Figure]

**Figure 9 Spatial and seasonal variability of the SMDI in trans-Himalaya (top), the mountains (middle), and the plains (bottom)**
Note: Each colour band shows the respective HRU's area combined.

[Figure]

**Figure 10 Spatial and seasonal variability of the SPI in the trans-Himalaya (top), the mountains (middle), and the plains (bottom)**

Note: Each colour band shows the respective HRU's area combined.

[Figure]

**Figure 12: Percentage of weeks with severe drought in the trans-Himalaya (top), the mountains (middle), and the plains (bottom )**
Note: Each colour band shows the respective HRU's area combined

---

## Author Response (AR2)

**Response to Editor**

Dear Editor,

Thank you for forwarding the comments from reviewer #2 who emphasised to compare the model-derived soil moisture with remote sensing products. In fact, we have considered the very approach while designing this study but did not include it in the main manuscript because of the following reasons:

1. Estimation of soil moisture from remotely sensed products is different from the model-based soil moisture assessment. For Example, NASA's Soil Moisture Active Passive (SMAP) estimates surface soil moisture within the top 5 cm of the soil and with a 2-3 day repeat cycle (Chan et al. 2016). Table 1 shows the available remote sensing-based soil moisture products. The J2000 model considers soil moisture for variable soil depths of up to 100 cm (depending on root depth of vegetation type) on a daily temporal resolution. The spatial resolution of SMAP is 1296 $km^2$ (36 km $\times$ 36 km) and 81 $km^2$ (9 km $\times$ 9 km) compared to the 4.7 $km^2$ average modelling unit size of the J2000 model (derived from 90-meter resolution datasets).

2. Remote sensing-based soil moisture estimates are also influenced by artificial water storage, surface irrigation and snow cover. In the plain areas of Koshi, both in Nepal and India, there is an extensive network of irrigation canals which supply water to irrigated lands. The soil moisture provided by irrigation systems would be different from the model-derived soil moisture. Similarly, in the high altitude area, snow cover can also affect the soil moisture signal.

3. Some of the recent remote sensing-based soil moisture products are available only after April 2015 (e.g. the SMAP satellite (Chan et al. 2016; Alemohammad et al. 2018)). We found Climate Change Initiative Soil Moisture product (CCI SM) by European Space Agency (ESA) which is available at 625 $km^2$ (25 km $\times$ 25 km) resolution from 1978 to 2015 (Dorigo et al. 2017). We have made a comparison between CCI SM and J2000 soil moisture for the period of 1980 to 2007 (Figure 1).

Figure 1 shows the monthly soil moisture comparison between CCI SM and the J2000 model. Because of their differences in soil depth considered, the comparison is made in a fraction of soil depth (i.e 40 cm for CCI SM and up to 100 cm for the J2000 model). The figure shows that both products illustrate the monthly variation in soil moisture where the soil moisture is high during the monsoon season and low in the spring season. However, the soil moisture volume difference is high. The CCI SM is about half in the trans-Himalaya and plains and one third in mountains compared to the J2000 model-derived soil moisture.

[Figure]

Figure 1: Comparision between soil moisture of CCI SM and J2000 model. First row: an average monthly comparison. Bottom row: scatter plot of monthly values (1980-2007).

As the remote sensing soil moisture is not a suitable basis for the validation of our model results, we opted not to show the comparison between model and remote sensing derived soil moisture in the main results, rather as a supplementary Figure of the comparison.

*To address the reviewer's comments, we have added the following lines in section 3.3 Hydrological modelling (last few sentences of the first paragraph) which highlights the limitation of these comparisons, and few sentences in 'uncertainty and limitation' section of 'Discussion' and provided the figure as Supplementary Figure 3.*

**Added portal in section 3.3 Hydrological modelling**

The soil moisture derived from the J2000 model could not be validated directly due to the lack of observed soil moisture data in the basin. While most of the remote sensing-based soil moisture is available only after 2015 (see e.g. Alemohammad et al. 2018), very few like the Climate Change Initiative Soil Moisture product (CCI SM) by European Space Agency (ESA) is available at 25 x 25 km resolution from 1978 to 2015 (Dorigo et al. 2017). Besides, these products differ in considered soil depth when compared to the J2000 model. The spatial resolution of the J2000 model is based on hydrological response units (HRUs) of an average size of 4.7 km$^2$, whereas all available satellite-based soil moisture products feature a distinctly lower spatial resolution. As an example, the CCI SM product has a spatial resolution of 625 km$^2$. Also, remote sensing products might capture artificial water storage, surface irrigation and snow cover, which also affect the spatial and temporal patterns of soil moisture. Because of these differences along with the J2000 model-derived soil moisture which typically considers root depth of vegetation which can reach up to 100 cm soil depth, direct comparison with satellite-derived soil moisture would not be reasonable in this study. However, a monthly comparison with CCI SM is provided in Supplementary Figure 3 and discussed in the 'Discussion' section.

**Added portion in Uncertainties and limitation**

We could not validate soil moisture result with station data due to lack of soil moisture network in the Koshi basin. Validation with remote sensing product was also not reasonable due to differences in soil moisture depth definitions and spatio-temporal resolutions. However, a comparison with CCI SM remote sensing-based soil moisture (Liu et al., 2011; Liu et al., 2012; Dorigo et al. 2017) suggests that both remote sensing and model shows inter-annual variability in soil moisture in which soil moisture is high during the monsoon season and low in the spring season but the absolute volume difference is high. The differences could be due to the different soil moisture depth in CCI SM (40 cm) and the J2000 model (up to 100 cm). Supplementary Figure 3 shows the comparison between CCI SM and J2000 model soil moisture comparison.

[Figure]

Supplementary Figure 3: Comparison between soil moisture of CCI SM and J2000 model. First row: an average monthly comparison. Bottom row: scatter plot of monthly values (1980-2007).

*Note about the figure:*

*CCI SM is a daily surface soil moisture, which has a spatial resolution of 25 km x 25km, as volume percentage for top 40 cm soil layer (Liu et al., 2011; Liu et al., 2012; Dorigo et al. 2017). We extracted the average monthly values separately for three regions. Because of differences in soil depth (i.e 40 cm for CCI SM and up to 100 cm for the J2000 model) in the compared datasets, the fraction volume of soil moisture for each product is presented. In average, CCI SM soil moisture is about half in the trans-Himalaya and plains and one third in mountains compared to the J2000 model.*

Table 1: Available remotely sensed soil moisture product

| Domain | Sensor and Resolution | Data available from | Reference |
|---|---|---|---|
| Global | • Soil Moisture Active Passive (SMAP) satellite
• 36 km resolution
• 5 cm of the soil column | March 31, 2015 and October 26, 2015 | Chan et al. 2016 |
| Global | • Soil Moisture Active Passive (SMAP) satellite
• 1 km (after downscaling), original resolution 36 and 9 km | April 2015 | Alemohammad, et al. (2018); Colliander et al. 2017 |
| Global | • **Soil** Moisture **and Ocean Salinity**

• 35–50 km | Satellite launched on 2 November 2009 | SMOS - Earth Online (esa.int) |
| Global | • *AMSR-E/Aqua L2B Surface Soil Moisture*
• 25 km resolution
• Surface soil moisture (up to 5 cm depth) | Temporal Coverage: 2002/06/18 to 2011/10/03 | Njoku et al. 2004 |
| Global | • *Climate change initiative Soil moisture by European Space Agency*
• *25 km resolution*
• *40 cm soil depth* | 1978–2015 | (Liu et al. 2011, 2012); Dorigo et al. 2017; ). |

**Reference**

Alemohammad, S. H., Kolassa, J., Prigent, C., Aires, F., & Gentine, P. (2018). Global downscaling of remotely sensed soil moisture using neural networks. *Hydrology and Earth System Sciences*, *22*(10), 5341-5356.

Chan, S.K., Bindlish, R., O'Neill, P.E., Njoku, E., Jackson, T., Colliander, A., Chen, F., Burgin, M., Dunbar, S., Piepmeier, J. and Yueh, S., 2016. Assessment of the SMAP passive soil moisture product. IEEE Transactions on Geoscience and Remote Sensing, 54(8), pp.4994-5007.

Colliander, A., Jackson, T. J., Bindlish, R., Chan, S., Das, N., Kim, S. B., Cosh, M. H., Dunbar, R. S., Dang, L., Pashaian…….., E. G., and Yueh, S.: Validation of SMAP surface soil moisture

products with core validation sites, Remote Sens. Environ., 191, 215–231, https://doi.org/10.1016/j.rse.2017.01.021, 2017b.

Njoku, E. G. 2004. *AMSR-E/Aqua L2B Surface Soil Moisture, Ancillary Parms, & QC EASE-Grids, Version 2*. [Indicate subset used]. Boulder, Colorado USA. NASA National Snow and Ice Data Center Distributed Active Archive Center. doi: https://doi.org/10.5067/AMSR-E/AE_LAND.002.

Dorigo, W., Wagner, W., Albergel, C., Albrecht, F., Balsamo, G., Brocca, L., Chung, D., Ertl, M., Forkel, M., Gruber, A. and Haas, E., 2017. ESA CCI Soil Moisture for improved Earth system understanding: State-of-the art and future directions. *Remote Sensing of Environment*, *203*, pp.185-215.

Liu, Y.Y., Dorigo, W.A., Parinussa, R.M., De Jeu, R.A.M., Wagner, W., McCabe, M.F., Evans, J.P., & Van Dijk, A.I.J.M. (2012). Trend-preserving blending of passive and active microwave soil moisture retrievals. Remote Sensing of Environment, 123, 280-297

Liu, Y.Y., Parinussa, R.M., Dorigo, W.A., De Jeu, R.A.M., Wagner, W., Van Dijk, A.I.J.M., McCabe, M.F., & Evans, J.P. (2011). Developing an improved soil moisture dataset by blending passive and active microwave satellite-based retrievals. Hydrology and Earth System Sciences, 15, 425-436